# Asymmetric peptidoglycan editing generates cell curvature in *Bdellovibrio* predatory bacteria

Emma J. Banks[1], Mauricio Valdivia-Delgado[2], Jacob Biboy[3], Amber Wilson[2], Ian T. Cadby [2,4], Waldemar Vollmer[3], Carey Lambert[1], Andrew L. Lovering [2✉] & R. Elizabeth Sockett [1✉]

Peptidoglycan hydrolases contribute to the generation of helical cell shape in *Campylobacter* and *Helicobacter* bacteria, while cytoskeletal or periskeletal proteins determine the curved, vibrioid cell shape of *Caulobacter* and *Vibrio*. Here, we identify a peptidoglycan hydrolase in the vibrioid-shaped predatory bacterium *Bdellovibrio bacteriovorus* which invades and replicates within the periplasm of Gram-negative prey bacteria. The protein, Bd1075, generates cell curvature in *B. bacteriovorus* by exerting LD-carboxypeptidase activity upon the predator cell wall as it grows inside spherical prey. Bd1075 localizes to the outer convex face of *B. bacteriovorus*; this asymmetric localization requires a nuclear transport factor 2-like (NTF2) domain at the protein C-terminus. We solve the crystal structure of Bd1075, which is monomeric with key differences to other LD-carboxypeptidases. Rod-shaped *Δbd1075* mutants invade prey more slowly than curved wild-type predators and stretch invaded prey from within. We therefore propose that the vibrioid shape of *B. bacteriovorus* contributes to predatory fitness.

---

[1] Medical School, School of Life Sciences, University of Nottingham, Queen's Medical Centre, Nottingham NG7 2UH, UK. [2] Institute for Microbiology and Infection, School of Biosciences, University of Birmingham, Birmingham B15 2TT, UK. [3] Center for Bacterial Cell Biology, Biosciences Institute, Newcastle University, Newcastle upon Tyne NE2 4AX, UK. [4] Present address: Bristol Veterinary School, University of Bristol, Langford, Bristol BS40 5DU, UK. ✉email: a.lovering@bham.ac.uk; liz.sockett@nottingham.ac.uk

*B*dellovibrio bacteriovorus HD100 is a small, vibrioid-shaped predatory bacterium that invades and then replicates within the periplasm of Gram-negative prey bacteria, forming a spherical structure called a prey bdelloplast[1]. *B. bacteriovorus* has a broad prey range which includes multidrug-resistant pathogens with variable outer membrane and cell wall chemistries, and the occurrence of genetic resistance to *B. bacteriovorus* has never been observed in prey bacteria[2,3]. Predatory *B. bacteriovorus* can also successfully clear pathogen infections within a range of in vivo animal models[4–6] and therefore has considerable and growing potential as a novel antibacterial therapeutic.

The predation process is critically dependent upon the modification of both predator and prey peptidoglycan (PG) cell walls to facilitate the dual bacterial encounter. PG forms a complex macromolecular structure called a sacculus which surrounds the cytoplasmic membrane of nearly all bacteria, maintaining cell shape and providing protection against lysis due to osmotic pressure fluctuations and large extracellular toxins[7]. Bacterial growth, cell division, and—importantly in this study—predation, occur through PG remodeling, which involves a repertoire of predator-secreted modifying enzymes[8–11].

The predatory lifecycle of *B. bacteriovorus* begins with attack-phase cells that swim[12] or glide[13] to encounter prey, then recognize and attach to the prey outer membrane. An entry porthole in the prey cell wall is created, through which the predator traverses to enter the inner periplasmic compartment[10]. Concurrently, two predator DD-endopeptidases are secreted into prey, cleaving crosslinks between prey PG peptide chains to sculpt rod-shaped prey cells into spherical bdelloplasts[8]. This also reduces the frequency of sequential predator invasions, thus conferring exclusivity to the first-entering predator[8]. The porthole in the wall and outer membrane is then resealed and the predator secretes hydrolytic enzymes including nucleases and proteases into the cytoplasm of the now-dead host, taking up the nutrient-rich degradative products[14,15]. Prey-derived and de-novo-synthesized nucleotides are incorporated into the replicating genome copies of the predator which grows as an elongating multi-nucleoid filament inside the rounded but intact prey until exhaustion of prey nutrients[16]. Synchronous septation of the predator filament yields progeny cells that secrete targeted PG hydrolytic enzymes to lyse the prey host and reinitiate the predatory cycle[11].

PG hydrolases have an additional role generally in the determination of cell shape[17], which has been particularly studied in the non-predatory, ε-proteobacteria *Helicobacter pylori*[18–20] and *Campylobacter jejuni*[21–23], in whom multiple PG hydrolases collectively generate helical morphology. In contrast, bacterial vibrioid morphology is generally determined by non-enzymatic cyto- or periskeletal proteins (well-studied in *Caulobacter crescentus*[24,25] and *Vibrio cholerae*[26,27]).

Despite the characterization of predator enzymes that modify the prey PG, there have been very few studies concerning the cell wall PG architecture or vibrioid cell shape of predatory bacteria. Here, we investigate the mechanism by which a curved, vibrioid predator is generated and ask whether there are evolutionary and functional connections between predator cell morphology and an efficient predatory lifestyle.

Here, we identify and characterize a predatory cell shape-determinant: Bd1075. Bd1075 is targeted to the outer convex cell face by its C-terminal nuclear transport factor 2-like (NTF2) domain, where it exerts localized LD-carboxypeptidase (LD-CPase) activity upon the *B. bacteriovorus* PG wall to generate curvature and the classical vibrio shape. We also show that the vibrioid cell shape of *B. bacteriovorus* predators facilitates rapid invasion into the periplasm of Gram-negative prey bacteria.

## Results

### Bd1075 generates the curvature of *B. bacteriovorus* predators.
The monocistronic *bd1075* gene of the vibrioid-shaped *B. bacteriovorus* Type strain HD100 encodes a 329 amino acid hypothetical protein with a predicted N-terminal sec signal peptide[28], suggestive of protein translocation into the periplasm or secretion from the cell (Supplementary Fig. 1). Bd1075 shares limited homology with Csd6 (identity: 24%, similarity: 38%) and Pgp2 (identity: 25%, similarity: 40%) which are dimeric proteins important for the generation of helical cell shape in *H. pylori*[18,29] and *C. jejuni*[21], respectively. These comparisons led us to hypothesize that Bd1075 could fulfil a role in the shape-determination of vibrioid predator *B. bacteriovorus*.

Reverse-transcriptase PCR (RT-PCR) revealed that *bd1075* is constitutively transcribed throughout the predatory cycle, suggesting that the protein may have a role in *B. bacteriovorus* rather than a secreted predatory function (Supplementary Fig. 2).

A markerless deletion of *bd1075* in the curved *B. bacteriovorus* Type strain HD100 could still be cultured predatorily (phenotype differences further detailed later) but Δ*bd1075* mutant cells had a distinct straight rod-shaped morphology unlike the curved wild-type HD100 parent strain (Fig. 1a, b). Wild-type median curvature (0.64 A.U., 95% CI [0.63, 0.66]) was significantly higher ($p < 0.0001$; Fig. 1c) than the Δ*bd1075* mutant (0.11 A.U., 95% CI [0.10, 0.12]). Plasmid-based complementation of Δ*bd1075* with the wild-type *bd1075*$_{HD100}$ gene increased curvature relative to the Δ*bd1075* mutant (Fig. 1c). These results indicate that Bd1075 has a role in generating the curvature of *B. bacteriovorus*.

The straight rod morphology of Δ*bd1075* resembles the long-cultured laboratory strain *B. bacteriovorus* 109J, which was isolated in the 1960s and is the only reported non-vibrioid strain of *B. bacteriovorus*[1]. As *bd1075* is conserved in all *B. bacteriovorus* strains including 109J and appears to be a curvature-determinant, we queried why strain 109J is non-vibrioid. Despite otherwise 100% sequence identity with *bd1075*$_{HD100}$, *bd1075*$_{109J}$ contains an in-frame N-terminal truncation of 57 amino acids (Pro-18 to Tyr-74) (Supplementary Fig. 3a-c). RT-PCR confirmed that *bd1075*$_{109J}$ is expressed in attack-phase cells and that the RNA transcript contains the predicted truncation (Supplementary Fig. 3d). To test whether the N-terminal truncation may render the translated protein non-functional in curvature-determination, we cross-expressed *bd1075*$_{109J}$ in the HD100 Δ*bd1075* mutant and found that this did not complement curvature (Supplementary Fig. 4). In contrast, cross-expression of *bd1075*$_{HD100}$ in wild-type 109J resulted in a median curvature of 0.29 A.U., 95% CI [0.26, 0.31], which is significantly higher than the curvature of wild-type strain 109J (0.17 A.U., 95% CI [0.15, 0.18], $p < 0.0001$; Supplementary Fig. 4).

These results show that Bd1075 is a curvature-determinant of vibrioid *B. bacteriovorus* strains and that an inactivating mutation within the gene resulted in the lab-evolved strain 109J, which is unable to generate cell curvature.

### Rod-shaped predators invade prey more slowly than the curved wild-type.
As cell morphology can be phenotypically important in other bacteria, we asked whether the curved shape of *B. bacteriovorus* could be advantageous to the bacterial predator during its unique intraperiplasmic lifecycle. Comparison of the gross predation efficiency of wild-type and Δ*bd1075* predators upon *E. coli* prey in either liquid culture, or on pre-grown *E. coli* biofilms, did not reveal a significant difference (Supplementary Fig. 6 and Supplementary Fig. 7). However, these are laboratory conditions with readily available prey and in which multiple important factors required to locate and navigate towards prey (e.g., predator chemotaxis and locomotion) are operational in bringing predators close to the prey surface.

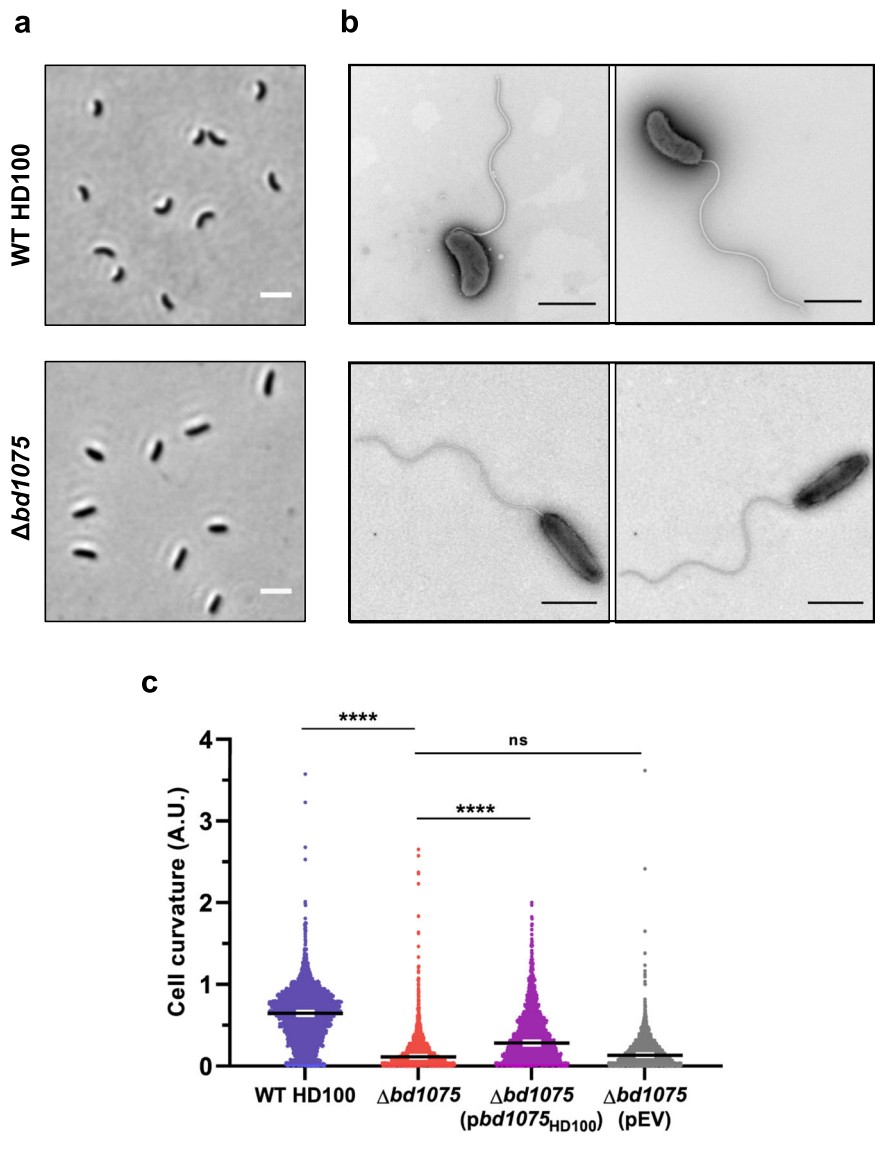

**Fig. 1 Bd1075 generates the curvature of *B. bacteriovorus* HD100 predator cells. a** Phase-contrast images of attack-phase *B. bacteriovorus* cells showing the curvature of wild-type (WT) HD100 cells in comparison to non-vibrioid Δ*bd1075* cells. Images are representative of cells from at least 5 biological repeats. Scale bars = 2 μm. **b** Transmission electron micrographs of WT HD100 and Δ*bd1075* cells stained with 0.5% uranyl acetate. Scale bars = 1 μm. Images are representative of 3 biological repeats. **c** Curvature measurements of *B. bacteriovorus* attack-phase cells. *n* = 2503 cells (WT HD100), 2149 cells (Δ*bd1075*), 1920 cells (Δ*bd1075* (p*bd1075*HD100)), or 2269 cells (Δ*bd1075* (pEV)) from 3 biological repeats. Error bars represent 95% confidence intervals of the median. ns non-significant (*p* > 0.05), ****\*p* < 0.0001; Kruskal–Wallis test with Dunn's multiple comparisons. Frequency distributions are included in Supplementary Fig. 5a. Source Data are provided as a Source Data file.

We considered that predator morphology may fulfill an important role at the interface of single predator-prey encounters and therefore studied predation more closely at the single-cell level using time-lapse microscopy to visualize individual predatory invasion events. *B. bacteriovorus* HD100 wild-type or Δ*bd1075* strains were mixed with *E. coli* S17-1 and placed under a microscope which captured images of specific fields of view every 1 min until the majority of *E. coli* prey had been invaded. Hypothesizing that curvature may affect the invasion of *B. bacteriovorus* into prey, we measured two parameters: prey attachment time and prey entry time (Fig. 2a). Duration of prey attachment did not significantly differ (*p* = 0.46) between the wild-type (median 28.5 min, 95% CI [28.0, 29.0]) and Δ*bd1075* (median 29.5 min, 95% CI [29.0, 30.0]) (Fig. 2b); however, there was a significant difference (*p* < 0.0001) between the rates at which the wild-type and Δ*bd1075* predators

entered prey: wild-type median 4.0 min (95% CI [4.0, 5.0]) versus Δ*bd1075* median 6.0 min (95% CI [5.0, 6.0]), respectively (Fig. 2c). Moreover, the longest wild-type entry was a single 7 min invasion, whereas 35.6% of Δ*bd1075* entry invasions were ≥7 min, with the longest invasion lasting 14 min. Complementation of Δ*bd1075* by single-crossover reintroduction of the *bd1075*HD100 gene into the genome of Δ*bd1075* restored entry time to wild-type levels (median 5.0 min, 95% CI [4.0, 5.0]; Fig. 2c).

These data indicate that *B. bacteriovorus* vibrioid morphology facilitates the traversal of predators across the prey cell envelope into the intraperiplasmic compartment of the rounded prey cell.

**Prey bdelloplasts are stretched and deformed by rod-shaped predators.** Having observed that the non-vibrioid Δ*bd1075*

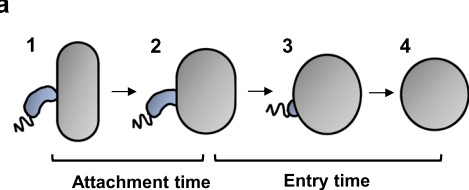

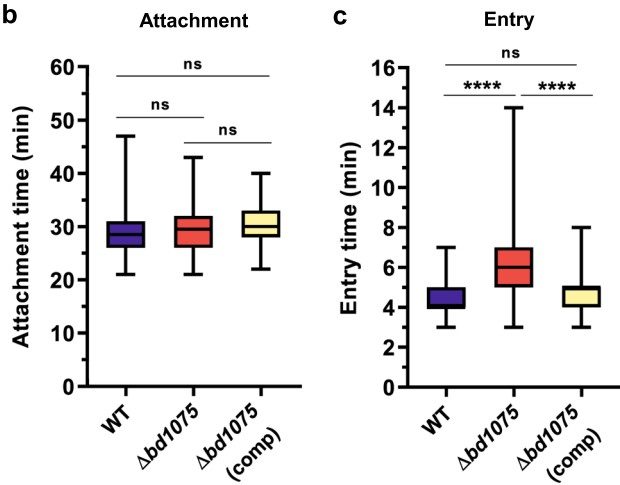

**Fig. 2 Prey attachment and entry times of *B. bacteriovorus* wild-type and Δ*bd1075*. a** Schematic to illustrate the measurement of attachment and entry times. Attachment time: number of frames (1 frame = 1 min) between initial predator attachment to prey and the first sign of predator entry into prey (stages 1–2). Entry time: number of frames between the first sign of predator entry and the predator residing completely inside the prey bdelloplast (stages 2–4). **b** Duration of attachment to and **c** entry into *E. coli* S17-1 prey by *B. bacteriovorus* HD100 wild-type (WT), Δ*bd1075*, and Δ*bd1075* complemented by single-crossover reintroduction of the *bd1075*_HD100 gene into the genome of the mutant: Δ*bd1075* (comp), measured by time-lapse microscopy. *n* = 90 cells, with 30 cells analyzed from each of 3 biological repeats. Box: 25th to 75th percentiles; whiskers: range min-max; box line: median; ns: non-significant (*p* > 0.05); *****p* < 0.0001; Kruskal–Wallis test with Dunn's multiple comparisons. Additional data plots are included in Supplementary Fig. 8, Supplementary Fig. 9, and Supplementary Fig. 10. Examples of *B. bacteriovorus* invasions are shown in Supplementary Fig. 11 and Supplementary Movies 1 (WT), 2 (Δ*bd1075*) and 3 (Δ*bd1075* (comp)). Source Data are provided as a Source Data file.

mutant was slower to enter the prey periplasm than the curved wild-type, we next investigated the growth of Δ*bd1075* within prey. During prey invasion, *B. bacteriovorus* secretes two DD-endopeptidases, Bd0816 and Bd3459, into the prey periplasm which hydrolyze the peptide bonds connecting chains of poly-saccharide backbone[8]. The prey PG wall becomes more malleable and the cell rounds up into a spherical bdelloplast.

We hypothesized that growth of the straight rod-shaped Δ*bd1075* within spherical bdelloplasts may be deleterious to the predatory niche, whereas curved wild-type cells may better 'fit' into the curvature of the bdelloplast during growth and elongation. A C-terminal mCerulean3 fusion to the continuously-expressed cytoplasmic protein Bd0064[6,30] was introduced via single-crossover recombination into both wild-type HD100 and Δ*bd1075* to label the predator cytoplasm blue and allow visualization of *B. bacteriovorus* within prey. Fluorescent *B. bacteriovorus* strains were mixed with *E. coli* S17-1 pZMR100 and observed throughout the predatory cycle. Wild-type predators elongated as tightly curved

filaments inside bdelloplasts (Fig. 3a); however, the rod-shaped Δ*bd1075* mutant—despite becoming more curved inside the spherical bdelloplast environment over time (Fig. 3b)—elongated as a less tightly curved filament and then septated to give rod-shaped, non-vibrioid progeny cells (Fig. 3a). Strikingly, a subset of Δ*bd1075* predator cells (~9.2%) appeared to stretch and deform the usually spherical prey bdelloplasts during intrabacterial growth (Fig. 3c).

Measuring the morphology of bdelloplasts containing a single *B. bacteriovorus* predator between 1 h and 2.5 h after predator-prey mixing showed that from 1 to 2 h, the shape of prey bdelloplasts did not obviously differ between the two strains. However, at 2.5 h when *B. bacteriovorus* cells are nearing maximal growth, the morphology of prey bdelloplasts became markedly different. The median area of each bdelloplast contain-ing a Δ*bd1075* mutant predator (1.74 μm², 95% CI [1.68, 1.79]) was significantly higher than bdelloplasts containing wild-type predators (1.63 μm², 95% CI [1.56, 1.73], *p* < 0.01; Fig. 3d) and the median circularity was significantly lower (Δ*bd1075*: 0.9925 A.U., 95% CI [0.9915, 0.9932]; wild-type: 0.9934 A.U., 95% CI [0.9931, 0.9951], *p* < 0.05; Fig. 3e). Bdelloplasts containing Δ*bd1075* predators were also significantly longer (median 1.55 μm, 95% CI [1.51, 1.58], *p* < 0.01; Fig. 3f) than those containing curved wild-type predators (1.51 μm, 95% CI [1.47, 1.55]) but the median width did not significantly differ (Δ*bd1075*: 1.37 μm, 95% CI [1.35, 1.39]; wild-type: 1.35 μm, 95% CI [1.32, 1.37], *p* > 0.05; Fig. 3g), consistent with the visual appearance of stretched bdelloplasts. These findings suggest that rod-shaped mutant predators can stretch and deform the spherical prey niche, in contrast to the curved wild-type.

**Bd1075 exerts LD-carboxypeptidase activity on PG sacculi in vivo and in vitro.** Bd1075 contains a predicted LD-transpeptidase (LDT) catalytic domain (Supplementary Fig. 1c, d); however, the LDT domains of related proteins Csd6 and Pgp2 function instead as LD-carboxypeptidases (LD-CPases)[18,21], which remove the terminal D-alanine from a PG tetrapeptide (consisting of L-Ala, D-Glu, *meso*-Dap, and D-Ala) to generate a tripeptide (consisting of L-Ala, D-Glu, and *meso*-Dap)[31]. This highlighted the need to verify the catalytic activity (if any) of Bd1075. PG sacculi were therefore purified from *B. bacteriovorus* strains and analyzed by HPLC to determine their muropeptide composition and any changes to it caused by Bd1075.

In contrast to curved wild-type HD100 sacculi, rod-shaped Δ*bd1075* sacculi contained a greater proportion of monomeric tetrapeptides (23.7 ± 0.8%) and crosslinked tetratetrapeptides (33.2 ± 0.7%) compared to the wild-type (9.6 ± 0.8% and 18.6 ± 0.6%, respectively), and a complete absence of mono-meric tripeptides and dimeric tetratripeptides (Fig. 4a, b and Table 1). This difference suggests that Bd1075 could cleave the C-terminal D-alanine of tetrapeptides to produce tripeptides which terminate with *meso*-Dap. The complemented strain Δ*bd1075* (p*bd1075*_HD100) contained no monomeric tetrapep-tides, 14.8 ± 1.2% tripeptides and 7.2 ± 0.5% dipeptides. These data suggest that reintroduction of the wild-type *bd1075*_HD100 gene resulted in over-complementation beyond wild-type as all monomeric tetrapeptides have been cleaved to tripeptides, with some subsequently converted to dipeptides (Fig. 4c and Table 1).

In comparison, the muropeptide profile of strain Δ*bd1075* (p*bd1075*_109J), in which curvature was not complemented (Supplementary Fig. 4), did not differ from Δ*bd1075*, further confirming the non-functionality of truncated Bd1075_109J as an LD-CPase (Supplementary Fig. 14a and Supplementary Table 1).

Wild-type 109J had a very similar muropeptide profile to the Δ*bd1075* mutant—a complete absence of tripeptides and

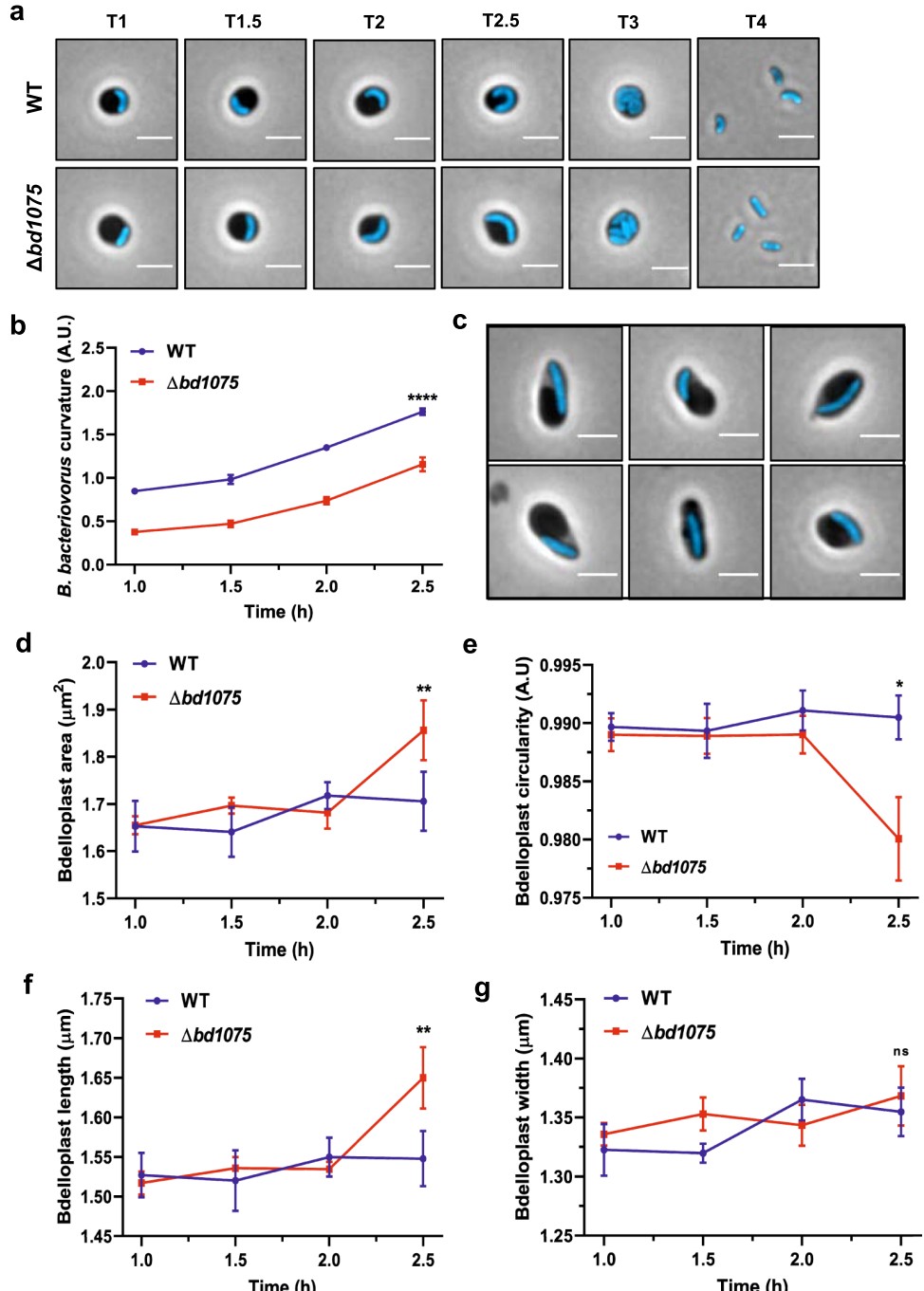

**Fig. 3 Intrabacterial growth and bdelloplast topology effects of *B. bacteriovorus* strains. a** Growth of *B. bacteriovorus* wild-type (WT) and Δ*bd1075* strains inside *E. coli* S17-1 pZMR100 prey bdelloplasts. *B. bacteriovorus* strains express the cytoplasmic fusion protein Bd0064-mCerulean3 to allow visualization of intraperiplasmic predator cells. T = hours elapsed since predators and prey were mixed. Scale bars = 2 μm. Images are representatives of cells from 3 biological repeats. **b** Curvature of *B. bacteriovorus* WT and Δ*bd1075* strains during predation upon *E. coli* S17-1 pZMR100 as depicted in **a**. Error bars represent standard error of the mean. ****p < 0.0001; two-tailed Mann–Whitney test. **c** Examples of Δ*bd1075* cells which appear to stretch and deform the *E. coli* prey bdelloplast at T = 2.5 h during 3 repeats of predatory timecourses as shown in **a**. These represented ~9.2% of total bdelloplasts. Scale bars = 2 μm. **d** Area, **e** circularity, **f** length, and **g** width of *E. coli* prey bdelloplasts during predation by WT or Δ*bd1075* predators. For data in **c–g**, n = 169 cells (1 h), 134 cells (1.5 h), 150 cells (2 h), and 160 cells (2.5 h) for the wild-type strain and n = 205 cells (1 h), 160 cells (1.5 h), 245 cells (2 h), and 250 cells (2.5 h) for Δ*bd1075* from 3 biological repeats. Error bars represent standard error of the mean. ns non-significant (p = 0.053); **p = 0.0039 (Area) or p = 0.0083 (Length), *p = 0.031; two-tailed Mann–Whitney test. Data presented with medians +95% CI and full data distributions are shown in Supplementary Figs. 12 and 13, respectively. Source Data are provided as a Source Data file.

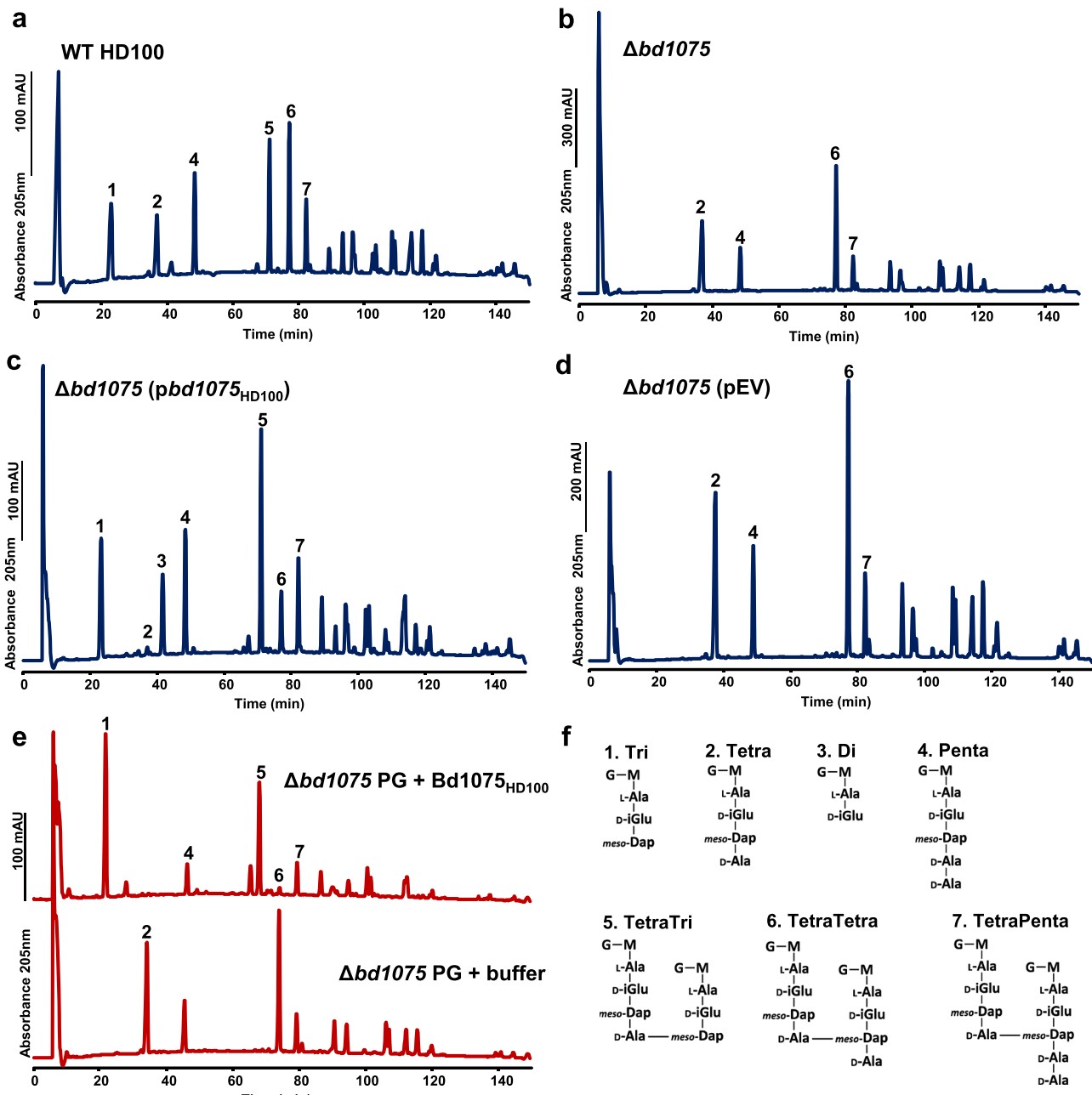

**Fig. 4 Muropeptide composition of *B. bacteriovorus* HD100. a–d** Muropeptide elution profiles obtained by HPLC. Peptidoglycan sacculi were isolated from attack-phase *B. bacteriovorus* cells. **a** Wild-type (WT) HD100, **b** Δ*bd1075*, **c** Δ*bd1075* (p*bd1075*HD100)—*bd1075*HD100 expressed in Δ*bd1075*, and **d** Δ*bd1075* (pEV)—empty vector control in Δ*bd1075*. Sacculi were digested by cellosyl and the resulting muropeptides were reduced with sodium borohydride and analyzed by HPLC. Representative chromatograms of 2 biological repeats are shown. **e** HPLC muropeptide elution profiles of Δ*bd1075* sacculi treated with either purified Bd1075HD100 enzyme (above) or buffer control (below). Data are from 1 biological repeat. **f** Structures of the seven main muropeptide fractions. Numbers correspond to those above peaks in **a–e** and were assigned based on the retention times of corresponding known *E. coli* muropeptides. G: *N*-acetylglucosamine, M: *N*-acetylmuramitol, L-Ala: L-alanine, D-iGlu: D-glutamic acid, *meso*-Dap: *meso*-diaminopimelic acid, D-Ala: D-alanine. Minor peaks are annotated in Supplementary Fig. 17 (for **a–d**) and Supplementary Fig. 16a (for **e**) and were assigned by mass spectrometry analysis (Supplementary Table 2).

tetratripeptides and a high proportion of monomeric tetrapeptides (28.0 ± 4.3%) and dimeric tetratetrapeptides (33.8 ± 1.1%) (Supplementary Fig. 14b and Supplementary Table 1). Finally, the cross-complementation strain 109J (p*bd1075*HD100), with an increased curvature compared to the lab-cultured wild-type strain 109J (Supplementary Fig. 4), contained a higher proportion of monomeric tripeptides (18.7 ± 3.0%) and dimeric tetratripeptides (26.2 ± 0.6%), and a reduction in monomeric tetrapeptides

dimeric tetratetrapeptides (3.0 ± 0.8% and 8.8 ± 0.8%, respectively) (Supplementary Fig. 14c and Supplementary Table 1). This demonstrates that cross-expression of *bd1075*HD100 in wild-type 109J resulted in the enzymatic conversion of tetrapeptides to tripeptides and increased the curvature of this normally non-vibrioid strain.

To further validate the LD-CPase activity of Bd1075, an N-terminally His-tagged copy of *bd1075*HD100 was expressed in *E.*

**Table 1 Quantification of muropeptides released from *B. bacteriovorus* HD100 sacculi.**

| Muropeptide | Relative peak area (%)[a] in *B. bacteriovorus* strain | | | |
|---|---|---|---|---|
| | WT HD100 | Δ*bd1075* | Δ*bd1075* (p*bd1075*$_{HD100}$) | Δ*bd1075* (pEV) |
| Monomers | | | | |
| Tri | 14.4 ± 1.0 | **n.d.\*[b]** | 14.8 ± 1.2 | **n.d.\*** |
| Tetra | 9.6 ± 0.8 | **23.7 ± 0.8\*** | 0.6 ± 0.9\* | **23.1 ± 2.7\*** |
| Di | 2.1 ± 0.1 | **n.d.\*** | 7.2 ± 0.5\* | **n.d.\*** |
| Penta | 12.8 ± 2.5 | 11.9 ± 3.6 | 12.2 ± 4.2 | 14.8 ± 0.0 |
| Monomer anhydroMurNAc | 2.2 ± 3.0 | 2.3 ± 3.2 | 2.6 ± 3.7 | 2.5 ± 3.6 |
| Monomers (Total) | 38.9 ± 0.5 | 35.6 ± 4.3 | 34.9 ± 2.6 | 37.8 ± 2.6 |
| Dimers | | | | |
| TetraTri | 14.9 ± 0.7 | **0.5 ± 0.7\*** | 26.4 ± 1.5\* | **n.d.\*** |
| TetraTetra | 18.6 ± 0.6 | **33.2 ± 0.7\*** | 7.8 ± 1.4\* | **29.6 ± 4.7\*** |
| TetraPenta | 12.2 ± 1.0 | 12.5 ± 0.3 | 14.7 ± 2.9 | 13.1 ± 0.2 |
| Dimer anhydroMurNAc | 17.3 ± 0.8 | 16.4 ± 0.6 | 20.5 ± 4.4 | 17.0 ± 0.3 |
| Dimers (total) | 45.6 ± 2.3 | 46.1 ± 0.6 | 48.9 ± 2.9 | 42.7 ± 4.5 |
| Trimers | | | | |
| TetraTetraTri | 1.6 ± 0.4 | **n.d.\*** | 3.8 ± 0.2\* | **n.d.\*** |
| TetraTetraTetra | 6.9 ± 0.5 | **11.2 ± 0.8\*** | 4.9 ± 1.0 | **10.6 ± 2.2\*** |
| TetraTetraPenta | 6.9 ± 3.6 | 7.1 ± 3.2 | 7.4 ± 4.3 | 8.8 ± 4.0 |
| Trimer anhydroMurNAc | 7.0 ± 0.3 | 9.5 ± 1.0\* | 6.2 ± 1.3 | 10.3 ± 1.0\* |
| Trimers (total) | 15.4 ± 2.8 | 18.3 ± 4.0 | 16.2 ± 5.5 | 19.4 ± 1.8 |
| Dipeptides (total) | 2.1 ± 0.1 | **n.d.\*** | 7.2 ± 0.5\* | **n.d.\*** |
| Tripeptides (total) | 22.4 ± 0.6 | **0.2 ± 0.3\*** | 29.3 ± 0.5\* | **n.d.\*** |
| Tetrapeptides (total) | 54.3 ± 1.1 | 79.3 ± 2.0\* | 41.4 ± 4.2 | 75.7 ± 1.4\* |
| Pentapeptides (total) | 19.0 ± 1.3 | 18.2 ± 0.9 | 19.5 ± 0.6 | 21.8 ± 5.0 |
| AnhydroMurNAc (total) | 13.1 ± 3.3 | 13.6 ± 2.6 | 14.9 ± 5.4 | 14.5 ± 3.8 |
| Average chain length | 7.9 ± 2.0 | 7.5 ± 1.4 | 7.2 ± 2.6 | 7.1 ± 1.9 |
| Degree of crosslinkage | 33.1 ± 0.7 | 35.3 ± 2.8 | 35.3 ± 2.2 | 34.3 ± 1.0 |
| % peptides in crosslinks | 61.1 ± 0.5 | 64.4 ± 4.3 | 65.1 ± 2.6 | 62.2 ± 2.6 |

Values represent the relative percentage area of each muropeptide peak in Fig. 4. Numbers with an asterisk differ from WT HD100 by more than 30% and values that are additionally emboldened differ by more than 50%. Source Data are provided as a Source Data file.
[a]Values are mean ± variation of two biological replicates.
[b]*n.d.* not detected.

*coli* BL21 and purified to near homogeneity by Ni-NTA affinity chromatography and size-exclusion chromatography (Supplementary Fig. 15). The muropeptide profile of *B. bacteriovorus* HD100 Δ*bd1075* sacculi incubated with purified Bd1075 enzyme revealed the complete conversion of both monomeric tetrapeptides to tripeptides and dimeric tetratetrapeptides to tetratripeptides (Fig. 4e). Bd1075 had identical enzymatic activity on wild-type 109J straight rod sacculi and upon sacculi of wild-type *E. coli* BW25113, showing that the enzyme can act on PG from different bacterial strains and species (Supplementary Fig. 16b-c).

These muropeptide data determine that Bd1075 has LD-CPase activity on PG both in vivo and in vitro, removing C-terminal D-alanine residues linked to the L-center of *meso*-Dap to convert tetrapeptides to tripeptides.

**Bd1075 structure determination.** The structure of mature Bd1075 protein was determined to 1.34 Å (Fig. 5 and Supplementary Table 3). The Bd1075 structure contains two domains: the catalytic LD-CPase domain (aa 47–180) and a C-terminal nuclear transport factor 2 (NTF2)-like domain (aa 196–304) (Fig. 5a). Interestingly, although there was a global agreement in the fold to Csd6 of *H. pylori* and Pgp2 of *C. jejuni* which also contain an NTF2 domain and an LD-CPase domain (the junction between the two at residue 188 of Bd1075), there were significant differences in fold elements and local regions. These differences resulted in an inability to solve Bd1075 via molecular replacement, necessitating the use of SAD phasing with co-crystallized halide ions. The Bd1075 protein is monomeric (the two molecules in the asymmetric unit contact one another by packing interactions only), lacking the dimerization regions of the other

characterized LD-CPase proteins; this was supported by size-exclusion data (Supplementary Fig. 18).

We were able to trace residues 29–308 with the exception of a presumably flexible region (aa 82–91) which we term the active site 'lid'. Differences to other LD-CPase structures are distributed throughout the fold (and in a small shift in NTF2:LD-CPase juxtaposition) as demonstrated by RMSD values for the full-length/LD-CPase-alone/NTF2-alone of 2.5 / 2 / 1.8 Å for Csd6 and 2.8 / 1.9 / 2.2 Å for Pgp2. The large values of 2.5/2.8 Å for full-length RMSD are in contrast with the agreement of 1.8 Å between Csd6 and Pgp2, hence Bd1075 is the structural outlier of the three proteins.

Bd1075 has a consensus active site, with the superfamily conserved catalytic triad consisting of C156, H141, and A142, each present in the expected (presumed active) orientations (Fig. 5b). A142 is often a glycine residue in other LD-CPases but here it makes identical h-bonding contacts to H141 using backbone carbonyl atoms. Bd1075 active site pocket-forming residue V158 is a relative anomaly as this position is arginine in most LD-transpeptidases or alanine in Csd6/Pgp2 and YafK-like enzymes shown recently to cleave the crosslink between PG and Braun's lipoprotein[32,33]. The termini of Bd1075 are very different to both Csd6 and Pgp2, replacing the N-terminal dimerization domain of those enzymes with a simple, shorter loop, and extending the C-terminus such that the Bd1075 NTF2 domain finishes with a longer beta-strand (aa 295–306)—residues of which contribute to its binding pocket (Fig. 5d, e). The C-terminus has a further 21 residues that we were unable to fit in this crystal form and which are predicted to be at least partly disordered.

Most noteworthy within the core of the Bd1075 C-terminus was the presence of W303, which is clearly located within the substrate-

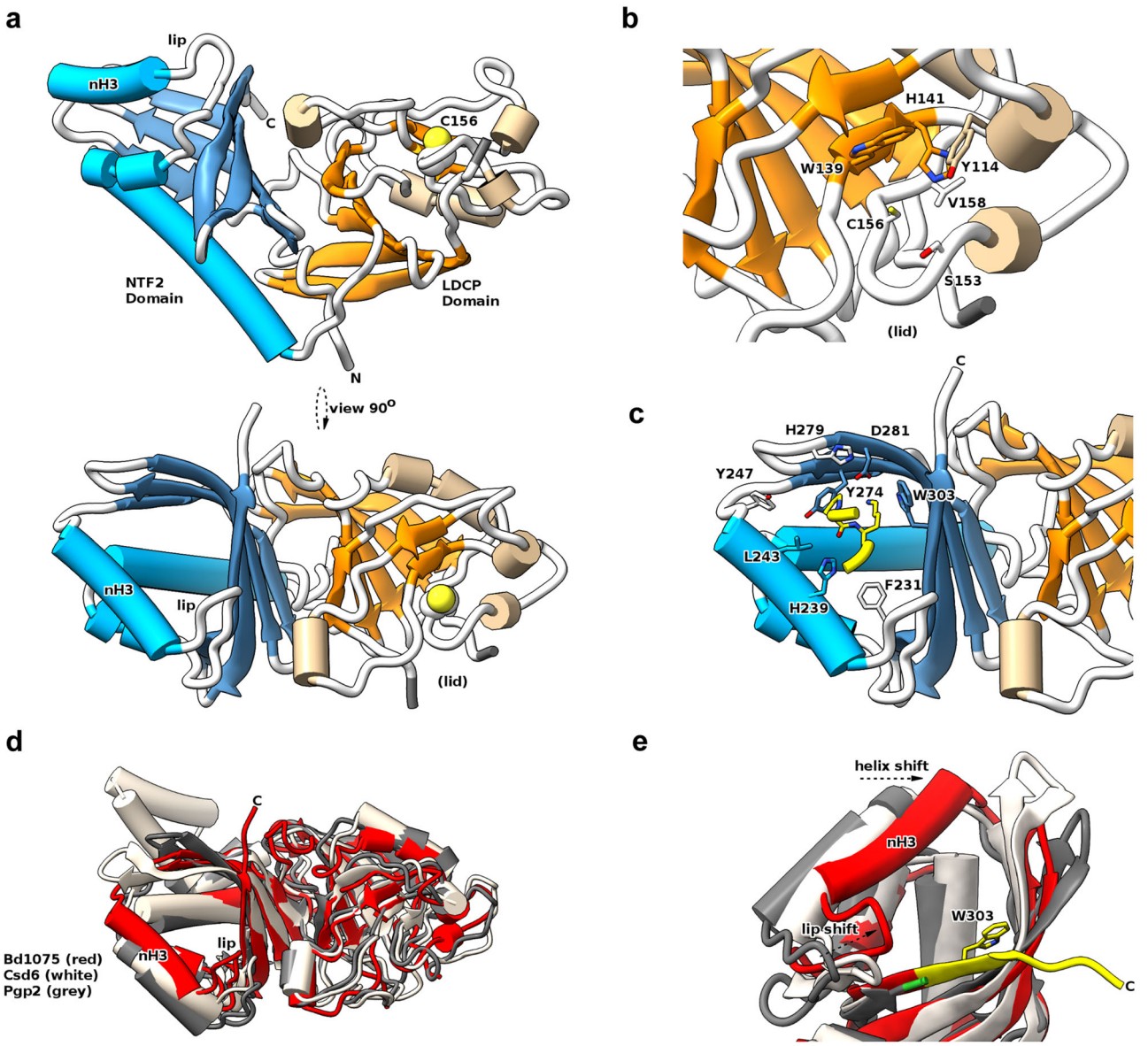

**Fig. 5 Structure of Bd1075 and features different to other characterized LD-CPase enzymes. a** Two orthogonal views of the Bd1075 fold, with catalytic residue C156 in space-fill form and features labeled. **b** Close-up view of the Bd1075 LD-CPase catalytic domain with selected residues that form the active site pocket displayed in stick form. **c** Close-up view of the Bd1075 NTF2 pocket, demonstrating complexation of a loop (residues 106–109 colored yellow, P107 and K108 in stick form) from a neighboring molecule in the crystal lattice. **d** Comparison of Bd1075 (red, 7O21: https://www.rcsb.org/structure/7O21), Csd6 (white, 4XZZ: https://www.rcsb.org/structure/4XZZ), and Pgp2 (gray, 6XJ6: https://www.rcsb.org/structure/6XJ6) structures. Helix 3 of the Bd1075 NTF2 domain (labeled 'nH3') and the associated loop ('lip') are relatively closer to the NTF2 pocket than the respective features of Csd6/Pgp2. **e** Close-up of the NTF2 terminus from structural alignment in **d**, demonstrating the relative extension of the Bd1075 C-terminus (colored yellow, includes NTF2 pocket-forming residue W303) in comparison to the shorter Csd6/Pgp2 termini (end residue colored green). The relative shifts of the nH3 helix and lip loop to constrict the NTF2 pocket are denoted by dashed arrows.

binding pocket of the NTF2 domain. W303 is highly conserved amongst *Bdellovibrio* Bd1075 protein homologs but not found in other LD-CPase proteins (due to their shorter NTF2 sequences which terminate at a position equivalent to Bd1075[E302]). The crystal packing of Bd1075 was such that a two amino acid loop of P107/K108 from one Bd1075 monomer packed into the NTF2 domain of the adjacent monomer. This loop is situated in an identical position to the bound glycerol of Csd6, postulated to be reminiscent of a substrate-like interaction[29]. Conserved residues of the Bd1075 NTF2 domain binding pocket pack around this feature (yellow in Fig. 5c), the base of which is formed by Y274—an important residue in both our monomeric structure and other dimeric LD-CPases.

Having begun to probe the cellular localization of Bd1075 in *B. bacteriovorus*, we used this new structural information to aid in the construction of fluorescently-tagged Bd1075 truncations and point mutants for enzyme localization tests.

**Bd1075 is targeted to the outer convex cell face by its NTF2 domain.** To determine whether Bd1075 is broadly active over all *B. bacteriovorus* envelope PG or if an activity is specifically localized, a double-crossover markerless strain in which mCherry is C-terminally fused to Bd1075 was constructed. Bd1075-mCherry localized to the outer convex face of *B. bacteriovorus*—both in free-swimming attack-phase cells (Fig. 6a) and throughout the predatory cycle (Supplementary Fig. 19). The periplasmic localization of

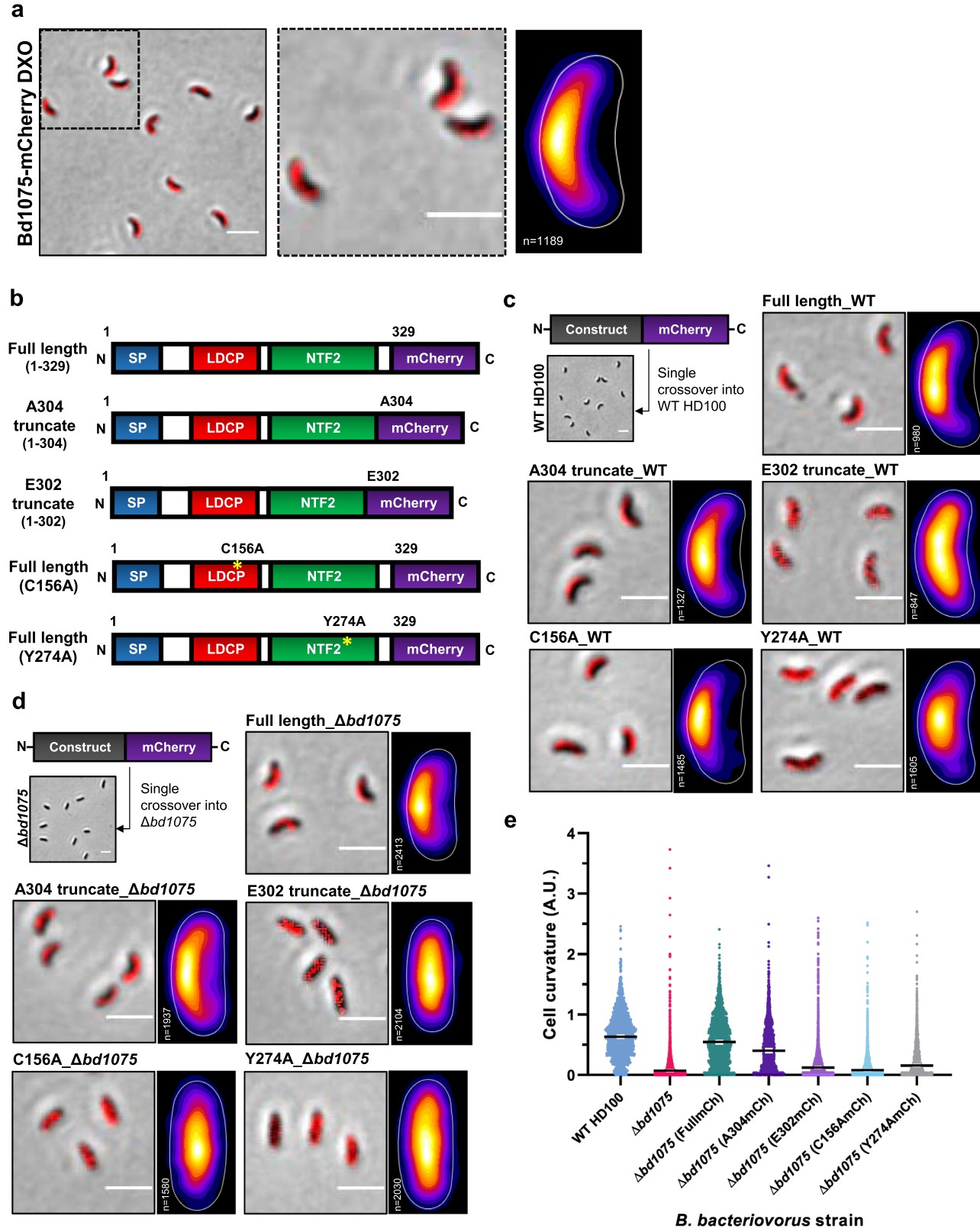

Bd1075 was tested via the construction of fluorescent fusions to mCitrine—a fluorophore incapable of periplasmic fluorescence. Bd1075-mCitrine gave no fluorescent signal in contrast to a Bd0064-mCitrine cytoplasmic control, suggesting that Bd1075 localizes to the periplasm (Supplementary Fig. 20).

Interestingly, unlike polar-localized proteins, the mechanism by which bacterial proteins are targeted to one of the cell lateral sides has not been described. Intrigued and partly informed by the protein structural data, we utilized the specifically targeted Bd1075 to investigate how bacterial proteins might be targeted to

**Fig. 6 The NTF2 domain is required to target Bd1075 to the convex cell face and generate curvature. a** *B. bacteriovorus* Bd1075-mCherry double-crossover (DXO) attack-phase cells (left), showing the localization of wild-type Bd1075-mCherry to the convex cell face, and representative of 3 biological repeats. Dashed boxed region is shown in a close-up (middle). Scale bars = 2 µm. Heatmap (right) depicts the location of wild-type Bd1075-mCherry foci detected in *n* = 1189 cells from 3 biological repeats. White-yellow = highest intensity, purple-black = lowest intensity. **b** Schematics of Bd1075-mCherry single-crossover constructs used in **c–e**. Full-length: Residues 1–329 (wild-type complete protein) fused to mCherry, A304: Residues 1–304 (contains a completed NTF2 domain including the *B. bacteriovorus*-specific residue W303) fused to mCherry, E302: Residues 1–302 (does not complete the NTF2 domain) fused to mCherry, C156A: Residues 1–329 (full-length with a point mutation of C156A in the catalytic LD-CPase domain) fused to mCherry, and Y274A: Residues 1–329 (full-length with a point mutation of Y274A in the NTF2 domain) fused to mCherry. **c**, **d** Bd1075-mCherry single-crossover constructs introduced into either *B. bacteriovorus* HD100 **c** wild-type (contains a native wild-type copy of *bd1075*) or **d** Δ*bd1075* (lacking a wild-type copy of *bd1075*). Attack-phase cell images and adjacent heatmaps show targeting of Bd1075-mCherry. Images and heatmaps were generated from 3 biological repeats (*n* = number of cells analyzed). Scale bars = 2 µm. **e** Curvature measurements of *B. bacteriovorus* Δ*bd1075* attack-phase cells containing different single-crossover Bd1075-mCherry fusions. *n* = 2099 cells (WT HD100), 1886 cells (Δ*bd1075*), 2577 cells (Δ*bd1075* (FullmCh)), 2170 cells (Δ*bd1075* (A304mCh)), 2812 cells (Δ*bd1075* (E302mCh)), 2083 cells (Δ*bd1075* (C156AmCh)), or 2523 cells (Δ*bd1075* (Y274AmCh)) per strain from 3 biological repeats. Error bars represent 95% confidence intervals of the median. All pairwise comparisons between strains (except for Δ*bd1075* vs Δ*bd1075* (C156AmCh)) were significant (*p* < 0.0001; Kruskal–Wallis test with Dunn's multiple comparisons). Frequency distributions are included in Supplementary Fig. 5c. Source Data are provided as a Source Data file.

just one lateral side-wall. We hypothesized that, in addition to the N-terminal signal peptide which targets the protein for translocation into the periplasm, Bd1075 may contain a second internal targeting sequence that directs the protein to just one lateral periplasmic side-wall. We therefore constructed five different protein variants of Bd1075, which were each fused to mCherry at the protein C-terminus (Fig. 6b). These constructs were then introduced into the *B. bacteriovorus* HD100 curved wild-type strain to generate single-crossover strains containing two copies of *bd1075*: the original wild-type and the new mCherry fusion. The correct and stable production of each fusion protein was confirmed by western blot analysis (Supplementary Fig. 21). The subcellular localization of each fusion protein was then examined by epifluorescence microscopy.

As expected, the full-length protein localized to the convex cell face (Fig. 6c) and we noted no morphological nor deleterious effects resulting from the presence of two functional copies of *bd1075*. The LD-CPase domain contains three conserved catalytic triad residues: His-141, Ala-142, and Cys-156. Mutation of Cys-156 to alanine (C156A) in full-length Bd1075 did not abrogate localization, indicating that LD-CPase activity is not involved in targeting (Fig. 6c).

As the Bd1075 C-terminal NTF2 domain (aa 196–304) is from a very broad protein superfamily[34], it was not possible to predict a putative function for this domain but the structure with the P107/K108 loop bound over the NTF2 pocket (Fig. 5c) suggested that this was a substrate interaction mimic and a means to destabilize putative PG substrate interactions. We tested a role for the NTF2 domain in protein targeting via three mCherry fusions: (1) Full-length Bd1075 containing an NTF2 domain point mutation changing Tyr-274 (which forms the base of the substrate-binding pocket, Fig. 5c) to alanine: Y274A; (2) Bd1075 truncated protein comprising residues 1-E302 (similar to the shorter Csd6 and Pgp2) which terminates 2 residues prior to the completion of the NTF2 domain at A304, omitting the highly conserved W303 which the crystal structure had suggested to be a Bd1075-unique feature; and (3) Bd1075 truncated protein comprising residues 1-A304, completing the NTF2-like domain and including W303.

The E302 mutant, which contains a truncated and incomplete NTF2 domain, failed to localize to the outer curve (Fig. 6c), while the Y274A fusion was capable of only partial localization (Fig. 6c), suggesting that this mutation in the NTF2 domain pocket may result in partial destabilization of the domain. In contrast, the A304 truncation mutant, which contains a complete NTF2 domain (and *Bdellovibrio*-specific residue W303), was correctly targeted to the outer curve (Fig. 6c).

These results strongly supported the hypothesis that the NTF2 domain is involved in protein targeting to the convex cell face and that the *Bdellovibrio*-specific NTF2 extension, including W303, is important for this function.

To investigate whether correct protein targeting is absolutely required to generate curvature, the five mCherry fusion constructs were introduced into the rod-shaped Δ*bd1075* mutant to generate single-crossover strains expressing solely the mCherry-tagged copy of *bd1075*. mCherry fusions were again confirmed by western blot (Supplementary Fig. 21) and the subcellular localization and curvature of each fusion strain were examined. The full-length protein localized to the outer curve and completely complemented the curvature of Δ*bd1075* (Fig. 6d, e). The LD-CPase catalytic domain point mutant C156A did not restore curvature and was diffuse throughout the rod-shaped cell (Fig. 6d, e). Critically, neither of the NTF2 domain mutants Y274A nor E302 were correctly targeted nor could either protein complement the curvature of Δ*bd1075* (Fig. 6d, e), highlighting the importance of the NTF2 domain and residue W303.

Together, these targeting data reveal that the NTF2 domain is responsible for targeting Bd1075 to the outer curve of the bacterial periplasm and that this specific localization is required to generate cell curvature.

## Discussion

In this work, we elucidate the vibrioid cell shape-determinant of predatory *Bdellovibrio bacteriovorus* bacteria and show that vibrioid morphology facilitates rapid prey invasion. These findings contribute to fundamental knowledge of bacterial cell shape and deepen our understanding of the predatory process, which may assist the application of predatory bacteria as a therapeutic.

In vibrioid bacteria, intermediate filament-like (IF-like) cyto- or periskeletal elements frequently determine cell shape[24,26]. IF-like proteins often contain coiled-coil rich protein (Ccrp) domains; however, deletion of the sole *B. bacteriovorus* Ccrp protein was previously found not to affect vibrioid cell shape[35]. Here, we discover that the vibrioid curvature of *B. bacteriovorus* is instead determined by a PG cell wall hydrolase: Bd1075. Refining the initial prediction of an LDT domain, we show via sacculus studies that Bd1075 functions as an LD-CPase, cleaving both crosslinked and uncross linked tetrapeptides to tripeptides in the predator PG cell wall (Fig. 4 and Table 1). This enzymatic activity was also observed for LD-CPase helical shape-determining proteins Csd6 (*H. pylori*) and Pgp2 (*C. jejuni*), highlighting the importance of biochemical validation of predicted LDT domains.

It is interesting to note that, while plasmid-based complementation of Δ*bd1075* with the wild-type *bd1075*HD100 gene resulted in partial complementation of cell curvature (Fig. 1c), the same strain showed 'over-complemented' enzymatic activity on PG, with complete conversion of tetrapeptides to tripeptides and subsequent cleavage to dipeptides (Fig. 4c and Table 1). A similar phenomenon was observed for the LD-CPase protein Csd6 and was caused by a 2.1× increase in protein production compared to the wild-type[18], whereby an excess of enzymatic activity resulted in 'under-complementation' of curvature. This suggests that an incorrect level of enzymatic activity leads to dysregulation of the system in general and the same is likely true here for Bd1075.

The overall peptide crosslinkage was slightly higher in cells lacking *bd1075* (64.4% compared to 61.1% in wild-type, Table 1). The reason for this effect is enigmatic but we hypothesize that a localized reduction in PG crosslinkage in wild-type cells could soften one lateral side-wall of the PG sacculus such that the cell bulges slightly and becomes deformed by internal cellular turgor pressure which pushes outwards to generate an outer convex curve that may be fixed by subsequent enzyme activity. A similar mechanism has been proposed to explain cell shape in *H. pylori* and *C. jejuni*[19–21,36].

*B. bacteriovorus* vibrioid curvature is widely conserved within this group of invasive predators with the exception of the rod-shaped and long-cultured laboratory strain *B. bacteriovorus* 109J. Originally named *B. bacteriovorus* 109, the strain was redesignated as 109J following the observation that, in one research laboratory, predator cells had transitioned from a curved to non-vibrioid shape[37]. It is possible that long-term laboratory culture conditions in which prey are highly abundant may have removed the selection pressure for vibrioid morphology, resulting in the lab-evolved 57-residue deletion that we detect to have inactivated Bd1075₁₀₉J. Bd1075₁₀₉J was not catalytically active upon PG (Supplementary Fig. 14a, Supplementary Fig. 16b, and Supplementary Table 1) and therefore could not complement curvature when cross-expressed in HD100 Δ*bd1075* (Supplementary Fig. 4). Despite retaining domains for LD-CPase catalytic residues and targeting capability, the 57-residue N-terminal truncation could severely disrupt Bd1075₁₀₉J protein folding and therefore function. The mutation most likely occurred via homologous recombination between 8 bp repeats which flank the deleted region (Supplementary Fig. 3b); this has been observed in other predator genes[38].

Bacterial morphology is evolutionarily conserved and known to confer selective advantages to different bacterial lifestyles[39,40]. The helical morphology of *H. pylori*, for example, facilitates efficient bacterial motility through the gastric mucosa to allow pathogenic colonization of the gastrointestinal tract[19]. Moreover, rod-shaped mutants of helical *C. jejuni* are deficient or have reduced fitness in a model of chick colonization[21,22]. In *V. cholerae*, vibrioid cell curvature (which is generated via a different non-enzymatic mechanism) increases motility through dense soft-agar matrices and promotes the colonization of *V. cholerae* in motility-dependent mice and rat infection models[26].

Here, we propose that *B. bacteriovorus* cell curvature facilitates efficient predatory invasion into Gram-negative prey bacteria. Straight rod-shaped *B. bacteriovorus* Δ*bd1075* predators invade prey significantly more slowly than curved wild-type predators (Fig. 2c). During prey invasion, *B. bacteriovorus* must presumably overcome opposing physical forces exerted upon itself by the turgid prey membrane and cell wall. Curved predators may distribute opposing forces as a glancing blow along the predator cell body, facilitating an efficient, curved trajectory into rounded prey, in contrast to force-intensive head-on invasions by rod-shaped predators.

Unlike wild-type curved *B. bacteriovorus*, non-vibrioid Δ*bd1075* predators stretch and deform the rounded prey bdelloplast (Fig. 3c–e). Intriguingly, non-vibrioid Δ*bd1075* predators become gradually curved during elongation inside spherical prey bdelloplasts, despite the absence of *bd1075* (Fig. 3a, b). The Δ*bd1075* predator does not curve as tightly as the wild-type, however, and released progeny cells are not curved but rod-shaped, indicating that adoption of an intracellular curvature is temporary in this mutant (Fig. 3a).

Various studies have shown that bacteria can sense and adapt their shape in response to different physiological topologies and adjust their internal physiology accordingly[41–43]. For example, *E. coli* cells that are subjected to mechanical strain within a confined environment can temporarily adopt the shape of that environment but then recover wild-type morphology upon release from confinement[44–46]. The mechanisms of shape-sensing and restoration of wild-type morphology are not yet fully understood, however[47].

Here, it is possible that in curved wild-type predators, Bd1075 may sense the temporary curvature imposed by the bdelloplast and enzymatically act on the *B. bacteriovorus* PG wall to tighten and permanently fix this curvature. This avoids potential damage to the replicative bdelloplast niche, while also preparing curved and invasively streamlined progeny predators for prey exit.

Fluorescently-tagged Bd1075 specifically localizes to the outer convex face of vibrioid *B. bacteriovorus* cells (Fig. 6a). Prior to this study, most cell shape enzymes (for which localization is known) were found at the inner concave face of bacteria. These include the *C. crescentus* cytoskeletal polymer crescentin[24] and *Vibrio cholerae* CrvA and CrvB which form periskeletal polymers[26,27]. Until the discovery of Bd1075, only the *H. pylori* bactofilin CcmA had been identified at the outer convex face, where it occupies the cytoplasm[36], unlike Bd1075 which is periplasmic (Supplementary Fig. 20).

To our knowledge, there is currently no known mechanistic basis for asymmetric bacterial protein targeting to one lateral side-wall of the cell. Intrigued by this and guided by the phenotyping of Bd1075 protein truncations and point mutations, we discover that the extended C-terminal NTF2 domain (including the unique pocket residue W303) targets the protein to the convex face and is necessary to generate cell curvature (Fig. 6c–e). NTF2 is a nuclear envelope protein that transports molecules into eukaryotic nuclei[48]. The NTF2-like domain superfamily, however, comprises hundreds of thousands of proteins spanning the 3 domains of life and is associated with over 200 biological pathways, suggestive of divergent evolution[34,49].

The NTF2 domain of *C. jejuni* Pgp2 was recently found by NMR studies to bind a variety of PG fragments, with specific secondary structure features shifting upon complexation[50]. The general agreement of some of the structures of monomeric Bd1075 with dimeric Pgp2 is suggestive that Bd1075 binds PG. The hydrogen-bond-rich nature of the P107/K108 loop we observe in our crystal structure could be indicative of such interaction by mimicking part of a PG muropeptide. Since the Pgp2 study invoked an induced fit on binding PG, it is particularly interesting that the third helix of the Bd1075 NTF2 domain (nH3, aa 233–248) and associated loop (labeled 'lip', aa 226–232) are shifted in relation to both Pgp2 and Csd6 (Fig. 5d). This helix was the major NTF2 feature that was reported to shift upon Pgp2-PG binding[50]. We postulate that its position has been modified in our Bd1075 structure by sidechains contacting the P107/K108 loop and may thus represent a 'bound' state. There is also the potential for NTF2:LD-CPase domain crosstalk, given that a small LD-CPase domain helix (aa 125–132) shifts in response to NTF2 alterations[50], and in Bd1075 appears to influence the disorder of the adjacent active site lid domain.

Since Bd1075 localizes to one lateral side-wall, one could theorize that the PG or outer membrane properties of that particular

side-wall must be uniquely different from the other. One possibility is that the NTF2 domain recognizes a modification or substrate which is more abundant at this side-wall. Alternatively, the NTF2 domain could sense the temporary physical curvature imposed by growth inside the spherical bdelloplast ('curvature-templating') and direct Bd1075 to this curved cell face. Bd1075 may then exert LD-CPase activity on the PG wall, initiating the fixation of *B. bacteriovorus* curvature. An alternative idea, which should not be excluded, is that the existing localization of Bd1075 at the convex face drives the targeting of newly synthesized Bd1075 to the convex face via self-interactions involving the NTF2 domain. These questions—which would present a significant investigative challenge beyond the scope of this study—could help to explain how bacteria both sense and respond to different topological space environments.

Collectively, these findings advance our understanding of factors affecting the physiology of therapeutically promising predatory bacteria and provide mechanistic insights into the evolutionary importance of bacterial cell morphology.

## Methods

**Bacterial strains and culture**. *Bdellovibrio bacteriovorus* strains were predatorily cultured in liquid Ca/HEPES buffer (5.94 g/l HEPES free acid, 0.284 g/l calcium chloride dihydrate, pH 7.6) and on solid YPSC overlays, both containing *Escherichia coli* S17-1 prey as described previously[51]. Where appropriate, kanamycin or gentamicin was added to growth media at concentrations of 50 or 5 μg ml$^{-1}$, respectively.

**Plasmid and strain construction**. Primers and plasmids used to construct strains used in this study are detailed in Supplementary Table 4 and Supplementary Table 5. Strains are listed in Supplementary Table 6. Bd1075 protein residue numbering was updated to correct for a previously mis-annotated start codon; the probable true start begins 2 codons downstream from that originally predicted (Supplementary Fig. 1b). To construct a markerless deletion of *bd1075*, 1 kb of upstream and downstream DNA were cloned into the suicide vector pK18*mobsacB* by Gibson assembly[52] using the NEBuilder HiFi DNA Assembly Cloning Kit (New England Biolabs). The vector was introduced into *B. bacteriovorus* via an *E. coli* S17-1 conjugal donor strain and a double-crossover deletion mutant was isolated by sucrose suicide counter-selection as described previously[8,53]. Deletion of *bd1075* was verified by RT-PCR, Sanger sequencing, and whole-genome sequencing.

Strains for complementation tests were constructed by inserting the *bd1075* gene from either strain HD100 or strain 109J (gene ID: EP01_15440) plus the respective native gene promoter into the vector pMQBAD, a derivative of pMQ414 which expresses the fluorescent tdTomato protein and is capable of autonomous replication in *B. bacteriovorus*[54]. Constructs for complementation tests were verified by Sanger sequencing, conjugated into *B. bacteriovorus* and maintained under a gentamicin selection pressure.

To initially fluorescently-tag Bd1075, the stop codon of *bd1075* was replaced with the *mCherry* coding sequence to generate an in-frame C-terminal fusion terminating at the stop codon of the mCherry protein. The fluorescent strain was constructed and verified analogously to the Δ*bd1075* mutant to generate a markerless double-crossover Bd1075-mCherry strain. For a more extensive analysis of domain functions, single-crossover mCherry fusions to either full-length or truncated versions of Bd1075 were made. These were constructed by cloning 1 kb of upstream DNA, the DNA encoding each desired Bd1075 domain-test (minus the stop codon), and the *mCherry* sequence into pK18*mobsacB*. Point mutations in such constructs were generated using Q5® Site-Directed Mutagenesis (New England Biolabs). The Bd1075-mCitrine fusion (Supplementary Fig. 20) was constructed in an analogous manner but replaced the *mCherry* sequence with *mCitrine*. Each construct was then conjugated into *B. bacteriovorus* to generate single-crossover merodiploid strains which were confirmed by Sanger sequencing and maintained under a kanamycin selection pressure.

**Protein expression and purification**. The *bd1075* gene from *B. bacteriovorus* HD100, minus the signal peptide and stop codon, was cloned into the vector pET41 in-frame with a C-terminal histidine tag and transformed into *E. coli* BL21. *E. coli* BL21 was cultured in TB media and incubated at 37 °C with orbital shaking at 200 rpm to an OD$_{600}$ of 0.6–0.8. *bd1075* expression was then induced with 0.5 mM IPTG at 18 °C for 16 h. Bd1075 was purified to near homogeneity using Ni-NTA affinity and size-exclusion chromatography, then dialyzed into either buffer containing 10 mM Na Citrate pH 6.0, 30 mM KCl, and 2% w/v glycerol (activity assays) or buffer containing 20 mM Na Citrate pH 6.0, 200 mM NaCl and 2 mM β-mercaptoethanol (structure studies).

**Muropeptide analysis**. To culture *B. bacteriovorus* for sacculi isolation, 1 l of each *B. bacteriovorus* strain was grown on either *E. coli* S17-1 or *E. coli* S17-1 pUC19 (gentamicin-resistant prey for complementation test strains containing the pMQBAD plasmid). Following complete predatory lysis of *E. coli* prey during a 24 h incubation at 29 °C with orbital shaking at 200 rpm, *B. bacteriovorus* attack-phase cultures were passed through a 0.45 μm filter to remove any remaining prey debris. To culture *E. coli* BW25113 for sacculi isolation, *E. coli* was grown at 37 °C for 16 h with orbital shaking at 200 rpm. Cultured *B. bacteriovorus* or *E. coli* cells were then centrifuged at 15,000 × g for 30 min at 4 °C, resuspended in 6 ml of ice-cold PBS, and then boiled in 6 ml of 8% SDS for 30 min to lyse the cells and liberate sacculi. Peptidoglycan was purified from the cell lysates, then muropeptides in the supernatant were reduced with sodium borohydride, and HPLC analysis performed as previously described[55]. The major muropeptides (peak no. 1–7) were assigned by comparing the elution times to those of well-characterized corresponding muropeptides from *E. coli* and other Gram-negative bacteria in the published literature[55,56]. The identity of the muropeptides with lower abundance and late elution times (peak no. 8–19) was confirmed by collecting and analyzing peaks from the WT HD100 chromatogram by mass spectrometry as described[57]. Mass spectrometry data are reported in Supplementary Table 2.

To test the in vitro activity of Bd1075$_{HD100}$, sacculi from either *B. bacteriovorus* HD100 Δ*bd1075*, *B. bacteriovorus* wild-type 109J or *E. coli* BW25113 were incubated with 10 μM of *B. bacteriovorus* Bd1075$_{HD100}$ in 50 mM Tris-HCl, 50 mM NaCl, pH 7.0 for 16 h at 37 °C on a thermomixer at 900 rpm. The control sample received no enzyme. To stop Bd1075 activity, the samples were boiled at 100 °C for 10 min and an equal volume of 80 mM sodium phosphate, pH 4.8, was added. The samples were incubated with 10 μg of cellosyl (Hoechst, Frankfurt am Main, Germany) for a further 16 h at 37 °C on a thermomixer at 900 rpm, boiled for 10 min and centrifuged at room temperature for 15 min at 16,000 × g. Muropeptides present in the supernatant were reduced with sodium borohydride, analyzed by HPLC[55] and identified as described above.

**Structure determination**. Purified Bd1075 at ~25 mg/ml was used for screening. Crystals were grown at 18 °C using the sitting drop technique in 4 μl drops composed of equal volumes of protein and reservoir solution. Bd1075 crystals were obtained in the BCS™ screen (Molecular Dimensions) in condition 2–44, comprising 0.1 M Tris pH 7.8, 0.1 M KSCN, 0.1 M NaBr and 25% PEG Smear broad range. Crystals were cryo-protected in mother liquor supplemented with 25% (v/v) ethylene glycol and flash cooled in liquid nitrogen. Diffraction data were collected at the Diamond Light Source in Oxford, UK (Supplementary Table 3). Data reduction and processing were completed using XDS[58] and the xia2 suite[59]. Bd1075 phasing was achieved using a merged SAD dataset (9000 frames, 0.1° oscillations) collected at a wavelength of 0.91 Å corresponding to the bromide anomalous scattering peak. The collected data were input into CCP4 online CRANK2[60], which located six bromide sites with an initial FOM of 0.14, followed by iterative cycles of building and model-based phasing improvement. The obtained model was further built and modified using COOT[61], with cycles of refinement in PHENIX[62].

**Phase-contrast and epifluorescence microscopy**. *B. bacteriovorus* cells were immobilized on a thin 1% Ca/HEPES buffer agarose pad and visualized under a Nikon Ti-E inverted epifluorescence microscope equipped with a Plan Apo ×100 Ph3 oil objective lens (NA: 1.45), an mCherry filter (excitation: 555 nm, emission: 620/60 nm), a CFP filter for mCerulean3 (excitation: 440 nm, emission: 470–490 nm) and a YFP filter for mCitrine (excitation: 508 nm, emission 540/25 nm). Images were acquired on an Andor Neo sCMOS camera with Nikon NIS software.

**Image analysis**. Images were analyzed using the Fiji distribution of ImageJ[63] and minimally processed using the sharpen and smooth tools, with adjustments to brightness and contrast. The MicrobeJ plug-in for ImageJ[64] was used to measure cell morphologies and detect fluorescent signal (foci). All images were manually inspected to ensure cells had been correctly detected before measurements were acquired. *B. bacteriovorus* attack-phase cells were generally identified by the parameters of area: 0.2–1.5 μm², length: 0.5–2.5 μm, width: 0.2–0.8 μm, and circularity: 0–0.9 A.U. To measure the curvature of *B. bacteriovorus* cells, MicrobeJ determines this as 'the reciprocal of the radius of curvature measured between the endpoints and the center of the medial axis of the cell' and it is described in arbitrary units (A.U.) (Supplementary Fig. 22). The fluorescent Bd1075-mCherry signal within attack-phase *B. bacteriovorus* cells was detected with the most appropriate MicrobeJ method (foci) using default maxima settings and an association with parent bacteria with a tolerance of 0.1 μm. Using this method, heatmaps could be generated to show the distribution of fluorescence signals across large numbers of cells. For the detection of Bd1075-mCherry fluorescent fusions in a curved wild-type genetic background, *B. bacteriovorus* cells were defined with the same parameters, but excluded cells with a curvature of <0.6 so as to only measure localization in cells with a definitively curved shape. Curvature parameters were set to 0-max to allow measurements of curvature for fluorescent fusions expressed in a non-curved Δ*bd1075* genetic background. To analyze prey bdelloplast morphology, bdelloplasts were generally identified by the parameters of area: 1.0-max μm², length: 0.5-max μm, width: 0.5-max μm, and circularity: 0.8–1.0 A.U. Circularity is defined in MicrobeJ as '4πr × area/perimeter²' with a value of 1.0 indicating a

perfect circle (Supplementary Fig. 22). Bdelloplasts with a circularity of <0.96 were classified as non-circular and stretched by Δ*bd1075* based on extensive visual observations and the fact that 0% of WT bdelloplasts had a circularity of <0.96 A.U. *B. bacteriovorus* cells were detected in the maxima channel with the bacteria method using default maxima settings and an association with parent bacteria with a tolerance of 0.1 μm. Only the morphologies of prey bdelloplasts which contained a single *B. bacteriovorus* predator were measured.

**Electron microscopy**. *B. bacteriovorus* cells were cultured for 24 h, then concentrated by microcentrifugation at $5000 \times g$ for 10 min followed by careful resuspension in 1 ml of Ca/HEPES. *B. bacteriovorus* cells were applied to glow-discharged Formvar/Carbon-coated 200-mesh copper grids (EM Resolutions), stained with 0.5% uranyl acetate for 1 min, then de-stained with Tris-EDTA pH 7.6 for 30 s. Samples were imaged under a FEI Tecnai G2 12 Biotwin transmission electron microscope at 100 kV.

**Time-lapse microscopy**. For time-lapse microscopy, 1 ml cultures of attack-phase *B. bacteriovorus* and 50 μl of stationary-phase *E. coli* S17-1 were microcentrifuged separately at $17,000 \times g$ for 2 min, then resuspended in 50 μl of Ca/HEPES. Predators and prey were then mixed together and immediately transferred to a thin 0.3% Ca/HEPES agarose pad. Cells were visualized under a Nikon Eclipse E600 upright microscope equipped with a ×100 oil objective lens (NA: 1.25) and a Prior Scientific H101A XYZ stage, which allowed six specific fields of view to be revisited over the time-lapse sequence. Image frames were captured every 1 min for at least 2 h on a Hammamatsu Orca ER Camera with Simple PCI software. Time-lapse videos of *B. bacteriovorus* attaching to and entering *E. coli* prey were analyzed in Simple PCI software. Attachment time was measured by counting the number of frames (1 frame = 1 min) between initial irreversible predator attachment to prey and the first indication of the predator moving into prey. Entry time was measured by counting the number of frames between the first indication of the predator moving into prey and the predator residing completely inside the bdelloplast.

**Reverse-transcriptase PCR**. To monitor the expression of *bd1075* across the predatory cycle, RNA template was isolated from different time points during a synchronous predation of *B. bacteriovorus* HD100 on *E. coli* S17-1 using an SV Total RNA Isolation System kit (Promega) as previously described[65]. RNA quality was verified on an Agilent Bioanalyzer using an Agilent RNA 6000 Nano Kit. RT-PCR was carried out using the QIAGEN OneStep RT-PCR kit with the following thermocycling parameters: 50 °C for 30 min, 94 °C for 15 min, followed by 30 cycles of 94 °C for 1 min, 50 °C for 1 min, and 72 °C for 1 min, and a final step of 72 °C for 10 min. Samples were run on a 2% agarose gel at 100 V for 30 min. Full, uncropped gel images are available in the Source Data file.

**B. bacteriovorus predation on E. coli in liquid culture**. The predation efficiency of *B. bacteriovorus* strains in liquid culture was measured using a luminescent prey assay developed by Lambert et al., 2003[66]. *B. bacteriovorus* strains were cultured in Ca/HEPES buffer at 29 °C with orbital shaking at 200 rpm for 24 h. Wild-type *B. bacteriovorus* HD100 and Δ*bd1075* predator strains were then matched by total protein content which was determined by Lowry assay[67]. Matched *B. bacteriovorus* strains were enumerated on overlay plates upon which plaques emerged after 5–7 days. Luminescent *E. coli* S17-1 was cultured in YT broth at 37 °C with orbital shaking at 200 rpm for 16 h. The stationary-phase *E. coli* was then adjusted to $OD_{600}$ 0.2 in a 1:1 mixture of Ca/HEPES and PY medium and 200 μl were aliquoted into each well of a 96-well microtiter optiplate. *E. coli* prey were enumerated on YT agar plates at 37 °C for 16 h and had a concentration of $10^7$ CFU ml$^{-1}$. One milliliter samples of *B. bacteriovorus* were heat-killed by incubation at 105 °C for 5 min. To create a predator dilution series, *B. bacteriovorus* live cell volumes of 0, 1, 2, 4, 8, 16, 32, and 64 μl were aliquoted into each well containing *E. coli* prey and made up to a total of 64 μl with heat-killed predator cells. Microtiter plates were covered with a Breathe-Easy® membrane and reduction in *E. coli* luminescence was measured at 30 min intervals in a BMG FluoStar microplate reader maintained at 29 °C with double orbital shaking at 200 rpm. To compare the rates of *E. coli* death by predation, the area under each luminescence curve was measured and then normalized to the maximum luminescence for each dilution. Data were analyzed in BMG LABTECH MARS data analysis software.

**B. bacteriovorus predation on E. coli biofilms**. The predation efficiency of *B. bacteriovorus* strains on prey biofilms was measured using an assay adapted from Lambert & Sockett, 2013[53] and Medina et al., 2008[68]. *E. coli* S17-1 was cultured in YT broth at 37 °C with orbital shaking at 200 rpm for 16 h and then 200 μl of stationary-phase *E. coli*, adjusted to $OD_{600}$ 0.1 in fresh YT, were aliquoted into each well of a 96-well PVC microtiter plate and incubated in a 29 °C static incubator for 24 h to produce a prey biofilm. *E. coli* prey were enumerated on YT agar plates at 37 °C for 16 h and typically had a concentration of $10^7$ CFU ml$^{-1}$. *B. bacteriovorus* WT HD100 and Δ*bd1075* were concurrently cultured at 29 °C with orbital shaking at 200 rpm for 24 h. Following complete prey lysis, *B. bacteriovorus* predators were filtered through a 0.45 μm membrane to remove residual prey. Predator strains were then matched by total protein and plaques were enumerated on double-layer overlay plates after 5–7 days. The *E. coli* biofilm-coated plate was removed from the incubator and carefully washed three times with Ca/HEPES buffer to remove residual planktonic cells. *B. bacteriovorus* 200 μl aliquots of neat ($10^0$), $10^{-1}$, $10^{-2}$, and $10^{-3}$ dilutions in Ca/HEPES buffer were then added to the *E. coli* biofilm plate. *B. bacteriovorus* filtered through a 0.22 μm membrane served as a no-predator control. The 96-well plate containing predators and prey was incubated for a further 24 h in a static 29 °C incubator. The plate was then stained with 1% crystal violet for 15 min, washed with sterile distilled water, and de-stained with 33% acetic acid for 15 min. Plate contents were mixed carefully and then absorbance readings were acquired at $OD_{600}$ to quantify the amount of *E. coli* biofilm remaining.

**Western blot analysis of Bd1075-mCherry fusions**. *B. bacteriovorus* Bd1075-mCherry fusion strains were cultured for 24 h until prey were completely lysed, filtered through a 0.45 μm membrane to remove residual prey debris, and then centrifuged at $5525 \times g$ for 20 min. The cell pellet was resuspended in 100 μl of 4× PAGE loading buffer containing β-mercaptoethanol, boiled for 5 min and then loaded onto a 4-20 % SDS-PAGE gel with either a MagicMark™ XP Western protein standard ladder (for blotting gel) or a SeeBlue™ Plus2 Pre-stained Protein Standard ladder (for loading control gel). Gels were run at 150 V for 1 h. Loading control gels were then stained and de-stained with QuickBlue Protein Stain (LubioScience). Blotting gels were transferred to a nitrocellulose membrane for 2 h at 25 V. Western blots used an anti-mCherry primary antibody (Invitrogen, product no: PA5-34974, diluted 1:4000) and a WesternBreeze™ Chemilumincescent kit according to the manufacturer's instructions. Images were captured by exposure to X-ray film.

**Statistical analysis**. Statistical analysis was performed in Prism 8.0 (GraphPad). Data were first tested for normality and then analyzed using the appropriate statistical test. The number of biological repeats carried out, n values for cell numbers, and the statistical test applied to the dataset are described in each figure legend.

**Reporting summary**. Further information on research design is available in the Nature Research Reporting Summary linked to this article.

## Data availability
The Bd1075 crystal structure data generated in this study have been deposited in the PDB database under the accession code 7O21. Source data are provided with this paper.

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

## Acknowledgements

This work was funded by a Wellcome Trust PhD studentship (215025/Z/18/Z) to E.J.B., a Becas Chile studentship (72180329) to M.V-D., a BBSRC studentship to A.W., and the UKRI Strategic Priorities Fund (https://www.ukri.org) EP/T002778/1 to W.V. R.E.S.,

C.L., and A.L.L. are currently funded by a Wellcome Trust Investigator Award in Science (209437/Z/17/Z). We thank Chloe Hudson for trials of Bd1075 purification, Daniela Vollmer for purification of PG, Joe Gray for mass spectrometry analysis of muropeptides, Rob Till for general laboratory support, and Block Allocation Group mx19880 for access to Diamond Synchrotron. Genome sequencing was provided by MicrobesNG (http://www.microbesng.uk), which is supported by the BBSRC (grant number BB/L024209/1). Electron microscopy was carried out at the Nanoscale and Microscale Research Centre at the University of Nottingham.

## Author contributions

E.J.B. made all deletion, complementation and fluorescent strains of Bd1075, carried out curvature analysis and conducted electron, time-lapse, and epifluorescence microscopy experiments with supervision by R.E.S. and C.L. J.B. carried out PG sacculi purifications and HPLC analysis with supervision by W.V. I.T.C. made the *bd1075* expression construct for protein purification. A.W. purified the protein with purification optimization and protocol design from I.T.C. M.V-D. crystallized and solved the structure of Bd1075. All protein work was supervised by A.L.L. E.J.B., R.E.S., and A.L.L. wrote the manuscript and all co-authors read and approved the final manuscript.

## Competing interests

The authors declare no competing interests.
