## [Peer Review File · Nature Communications]

Reviewers' Comments:

Reviewer #1:

Remarks to the Author:

The paper by Banks et al. reports on a novel mechanism of curved shape generation in *Bdellovibrio*. This is a lovely paper, well-written and easy to follow, with well-controlled and carefully interpreted experiments reporting a significant advance. I only have a few minor suggests/comments.

1. Please supply movies underlying the analyses for Fig. 2 as supplementary material (which is currently marked as "Supplementary Information is available at [x])
2. I am intrigued by their model of how bd1075 contributes to curvature (particularly the chicken vs. egg problem that arises). I like the idea that mechanical stress might produce some degree of curvature (which is also consistent with their data in Fig. 3B showing a time-dependent increase in curvature even in the mutant), which is then re-inforced by PG modifications. Some observations in *E. coli* (PMID 28737752) seem to suggest that this is in principle possible, and it might be useful to incorporate this paper into their discussion.
3. Regarding the discussion in line 484 – enhanced L,D CPase activity might actually promote L,D crosslinking (rather than reduce overall crosslinking as stated) by supplying more tripeptide substrate for L,D TPases, or rather induce a shift from DD to L,D crosslinks (and this might in some way promote curvature). Did you look specifically at 3-3 crosslink levels in the mutant by any chance? This effect might be too local to be globally detectable though.

Reviewer #2:

Remarks to the Author:

This manuscript by Banks et al. presents a study on the role of the protein Bd1075 has on determining the morphology of the vibrioid predatory bacterium *Bdellovibrio bacteriovorus*. Sequence alignment studies suggests that the hypothetical protein is comprised of two domains, an N-terminal L,D-carboxypeptidase (LDCP) domain and a C-terminal NTF2 domain of unknown function. Experiments are described that demonstrate the activity of Bd1075 as an LD-carboxypeptidase, its crystal structure, and the role of the NTF2 domain in localizing the enzyme to the outer curved face of *B. bacteriovorus*. The authors conclude that Bd1075 is indeed responsible for curvature of the bacterium and that it contributes to the overall efficiency and fitness of this bacterial predator.

The study is novel and provides important insight regarding structural determinants for bacterial cell morphology. The manuscript was prepared with great care and attention to detail while being concise; it is clear and easy to read. This reviewer has only a few relatively minor issues that the authors should address to enhance the manuscript.

1. Lines 92-110: This text of the introduction re-iterates much of the Abstract as it reports on the findings of the current work. As such, it could be abbreviated significantly.
2. Line 157: The results presented in the text above suggest that Bd1075 contributes to the curvature morphology but not necessarily the (only) determinant. For example, it cannot be discounted at this juncture that Bd1075 works in concert with one or more other proteins/enzymes for this curvature.
3. Line 300: The authors need to provide the evidence for the purification of recombinant Bd1075 to near homogeneity. Perhaps this could be accomplished with the presentation of SDS PAGE analysis together with the SEC result presented in Supplemental Fig. 11 (i.e., add the SDS PAGE analysis now making it Supplemental Fig. 9).
4. Legend to Figure 4a-d: The authors need to make it clear to the uninformed reader that the "reduced muropeptides" were "released" from digested samples of isolated peptidoglycan; as it reads now, the reader might think the muropeptides had been naturally released.

5. Legend to Figure 4f: Were these structures confirmed by MS analyses or simply assigned based on elution times and comparison to published studies; the Methods are silent on the use of MS? This should be made clear here and in the legends to Supplemental Figs 8–10 as well.

6. Table 1: The table legend states that the values represent the “relative percentage area” of each muuropeptide peak in Fig. 4. The Methods section for this experiment simply refers to Reference 48, but were the appropriate response factors for the various muuropeptide oligomers used for these quantifications? If so, this should be made more clear here and stated in the Methods as the data are reporting more than simple area under the curves for each identified HPLC fraction. If not, and simple areas are indeed reported, then they should be manipulated by the appropriate response factors to provide a more accurate comparison of the muuropeptide fractions. This applies here and also to Supplemental Table 1.

Reviewer #3:

Remarks to the Author:

Review Banks et al

In this manuscript the laboratories of Dr. Sockett and Dr. Lovering study the role of Bd1075 in generating cell shape in the predator bacteria *B. bacteriovorus*. Using a wide collection of methods, their main findings are that Bd1075 is:

- 1 - important for generating vibroid morphology
- 2 - important for optimal predation
- 3 - a LD-Cpase, and that its enzymatic function is important for vibroid morphology
- 4 - localizes asymmetrically via its NFT domain, which is also important for vibroid morphology

This is a nicely constructed, and reported study. I have just a few small issues:

1- The complementation results are a little confusing. Most of their figures show only partial complementation of the vibroid shape by expressing the wildtype allele (Figure 6e is different and it seems like the mCherry fusion complements more?). However, their mass spec data show that the complementing strains may be ‘overly complementing’, in that the complemented strain has no monomeric tetrapeptides. I think it would be worth discussing this discrepancy somewhere in the discussion.

2- For the single cell data, the replicates are all shown together, and a p-value is calculated between the averages of the strains (a very high N, giving a very small p-value). However, it is more rigorous to separate out the replicate data (this can be done as a ‘super-plot’) and calculate a p-value comparing the average of the averages (so an effective N of 3).

Reviewer #4:

Remarks to the Author:

Banks et al. report that the Bd1075 protein is involved in setting *Bdellovibrio bacteriovorus* cell curvature. The authors state that the vibrioid morphology increases the evolutionary fitness of predatory bacteria, improving prey invasion and intracellular growth. They investigate the protein domains responsible for its intriguing localisation to the convex side of the cell. The authors also demonstrate that Bd1705 is an LD-carboxypeptidase and solved the structure of the Bd1705, revealing interesting features compared to other related PG-modifying enzymes. Overall, although potentially important not only for the *Bdellovibrio* research field but also for increasing our knowledge of bacterial shape determinants and their physiological roles, this study raises several concerns. While the data clearly indicate a role of Bd1075 in cell curvature, interesting structural features and its LD-carboxypeptidase activity, several major conclusions are not fully supported by data in the present state and, in my opinion, require additional controls and stronger statistical analysis.

Major comments

I have 3 major concerns about this manuscript.

1) My biggest concern relates to the way most data from in vivo experiments are analysed.

- At several instances, the mean is inappropriately used as a statistical value to compare populations that show non-normal distributions (methods indicate that normality was verified but this result is very surprising considering data shown in Supplementary Figure 5). Besides, means are sometimes calculated from several replicates, but the distribution of all data points indicates quite some variability between repeats (Supplementary Figure 7), questioning the relevance of the mean as a proxy for statistical comparison of conditions/strains. Along the same line, some figures (e.g. Figure 2b-c, Figure 4b-c) presenting means calculated from several replicates would benefit from showing all data points or full distributions and their source (i.e. which replicate) instead of the single mean value only.

- It is unclear how the authors chose particular statistical tests to compare means in different figures (e.g. Kruskal-Wallis test for Figure 1c, Mann-Whitney test in other figures), and whether the chosen tests are the most appropriate in cases of non-normal distributions.

- The authors should clarify or modify their analysis of the prey-death curves in Supplementary figure 6, which aims at comparing the predation efficiency of the Δ bd1705 mutant with the wild-type strain. Regarding the negative predation control in panel a: heat-killed predators were added, so no prey death is expected. Could the authors explain why they see a decrease of luminescence in a, and whether the control values were taken into account in panel b? Why using the area under the curve as a comparison parameter, especially considering that the test curves are overlapping with the negative control curve for half of the time? Calculating the slope of each prey-death curve would seem more straightforward in my opinion. Is the luminescence increase until 4h due to prey growth/proliferation? It is unclear what each data point represents in panel b and how this type of calculation supports the absence of difference between the two strains. Points seem quite spread around the linear fit. Is fitting not affected by the precision limit of plaque counts?

- Data in Figure 3 do not convincingly support the conclusion about vibrioid shape being a fitness advantage for *B. bacteriovorus*. The frequency of the clear cases of bdelloplast elongation (panel c) should be calculated and compared between wt and Δ 1705 strains. The authors only refer to a "sub-set" at Line 221. Could this sub-set be responsible for the changes in mean bdelloplast dimension shown in panels d-g (mean values being sensitive to low numbers of outliers)? The authors could strengthen their analysis by testing if the 109J strain significantly deforms the bdelloplast from within. I am also puzzled by the apparent absence of difference in width (panel g), although bdelloplasts seem narrow in panel c, and related to this, by the apparent increase in bdelloplast area (panel d). How could the area of the bdelloplast change over time? I would expect this parameter be fixed by the initial dimensions of the prey. I am also not convinced that the statistical significance obtained in panel e (mean \pm SD circularity values: 0.98 ± 0.04 vs 0.99 ± 0.01) relates to a physiological difference in shape.

2) My second concern relates to the interpretation of the sub-cellular localisation data.

- Supplementary figure 12. It is confusing that the Bd0064-mCerulean signal is thinner and shorter than the Bd1705-mCherry signal in the corresponding cells (where the opposite is expected). This might come from (too) strong contrasting of the Bd0064 images (and additional processing mentioned in the methods). I suggest that the authors refine their image display.

- The authors claim at several places that the predicted N-terminal signal sequence targets Bd1705 to the periplasm. Periplasmic localisation of the protein is however not shown. This is an important missing piece of data considering that the authors conclude on the cell wall modification role of this protein.

- I have several concerns related to the interpretation of data in figure 6c-d, where Bd1705 variants are produced in fusion with mCherry, either in presence of the wild-type untagged Bd1705

copy or in a bd1705 null strain.

(i) The authors do not fully consider the localisation patterns of the variants in the Δ bd1705 strain (shown in d) to conclude on the impact of certain residues and domains on the localisation of Bd1705. Yet it seems that the wild-type copy impacts their fusions, possibly in different manners than suggested. For instance, they claim that the C156 of the catalytic domain is not required for localisation (lines 413-415), but the same mutant is unable to localise at the convex side in the absence of the wild-type Bd1705, suggesting that C156 does have an impact on the subcellular localisation, and that it localises at the convex side only if the wild-type protein is there. While the authors propose that this is due to curvature sensing by the C156A mutated form (lines 580-585), it is also possible that it is simply recruited by the wild-type form. They have not excluded the possibility of self-interactions in *B. bacteriovorus* in vivo and therefore the claims from this figure should be modified .

(ii) The heat map of Y274A in panel c seems to indicate an intermediate pattern (between the E302 mutant and WT), which is not commented (the authors conclude on the same inability to localise at convex sites for both E302 and Y274A mutants, which is the case in the Δ bd1705 but not in the wt background). The conclusions drawn from these data (i.e. that the C-terminal NTF2 extension is important for localisation) should be reconsidered: the NTF2 domain is important for localisation but a residue in the catalytic domain as well.

(iii) It is crucial to confirm by western blot that all fusions in all backgrounds are properly produced and not truncated (which would affect the distribution of the fluorescent signal).

(iv) The authors state that the C156A-mutated fusion that do not display asymmetric localisation remains localised to the cytoplasm. The fluorescent signal seems indeed to fill the whole cell. How would this mutation impact the periplasmic targeting of the protein?

3) Several main claims are not supported by data and should be toned down.

- Based on data in Figure 2, the authors draw the important conclusion that entry time is longer for the mutant than for the wild-type. (i) Complementation is needed to support the claim, especially since the strain is available. (ii) The time difference being very small (4.3 ± 0.9 min vs 6.1 ± 1.7 min) and using means (despite an apparent non-normal distribution for Δ bd1705), I am not fully convinced that it is physiologically relevant (how does it compare with the time-resolution of the experiment / with the variability of this event?). (iii) The authors then go further in their conclusion by saying that these data, combined with the curvature differences observed in Figure 1, indicate that the vibrioid morphology facilitates the traversal of predators across the prey cell envelope into the periplasmic compartment of the rounded prey cell. This claim should be toned down, since they have not shown directly that the curvature defect in cells lacking bd1705 is what causes the reported entry delay (in addition to the apparent weakness of these data in their present form).

- The claim that the vibrioid morphology confers optimal intracellular growth (e.g. line 468) is overstated. The data do not show that the modified bdelloplast shape and the lower curvature of the predator cells impact their growth per se. For this, analysis of growth rates or number of predator cells released per bdelloplast would be needed. The absence of obvious predation efficiency difference between the WT and the Δ bd1705 cells suggests instead that there is no major benefit for growth conferred by curvature. This claim about optimal growth / evolutionary advantage provided by curvature should be softened.

- The claim that the C-terminal NTF2 domain targets the protein is not fully supported by the data, showing that a mutation in the catalytic domain also prevents the localisation of the Bd1705 protein fusion to the convex lateral side (see major comment #2).

- At several places in the manuscript, the authors write "what becomes" the convex cell face (e.g. lines 385, 410, 434, 442, 542). However, at this point, the authors have not demonstrated that the proposed asymmetric localisation of Bd1705 arises before the establishment of curvature. This would require inducing the expression of a Bd1705 fluorescent fusion over time in a bd1705-null background (straight cells) and the observation of curvature increase after asymmetric positioning of Bd1705. Using "what becomes the convex face" might also sound counter-intuitive, since the mother *B. bacteriovorus* cell is already curved before it starts to grow. Please clarify or modify

throughout.

Minor comments

- Discussion, lines 531-537 / 585-590: The idea of topological sensing the physical space to prime asymmetric curvature (and therefore curvature-mediated asymmetric protein localisation) is interesting. However, the idea of topological sensing of the physical space and its impact of bacterial morphology is not a new concept, unlike what the text seems to suggest (see for instance Männik et al, PNAS 2012; Wu et al, Nat Nanotech 2015; Söderström et al, Nat Commun 2018). Curvature-mediated localisation has also been described for many proteins in various species. Besides, I am also thinking of alternative explanations for the asymmetric localisation of Bd1705. In fact, it is not clear why Bd1705 would help fixing a bdelloplast-induced curve. Is there evidence that the growing *B. bacteriovorus* always "sticks" to the bdelloplast membrane? I would assume that the filament is sometimes twisting in the middle of the bdelloplast, where curvature cannot be imposed by the bdelloplast boundaries. How about the curvature of free-living *B. bacteriovorus* variants? If these are curved, is their shape modified in the absence of bd1705? If yes, that would suggest that there is no need for a bdelloplast-induced curve at the first place, and that Bd1705 induces curvature (directly or not) independently of the physical constraints of a spherical and confined environment ("topological sensing"). A perhaps more straightforward interpretation of the asymmetric localisation of Bd1705 would simply arise from the presence of Bd1705 in the mother cell, which could recruit and direct the localisation of newly synthesised Bd1705 on that side in the growing *B. bacteriovorus*. This alternative hypothesis would be in line with the idea that self-interaction and recruitment, rather than topological sensing, might determine asymmetric localization of Bd1705 in *B. bacteriovorus*.

- The authors write that most known cell shape determinants are localised at the inner concave face of bacteria. Are there many more of those beside crescentin and CrvA/B? If not, then the convex localisation of Bd1705 and CcmA is just as frequent (2 cases concave vs 2 cases convex).

- The order of appearance of supplementary figure references in the text is generally not following the order of these supplementary figures. Please reorder supplementary figures and/or panels.

- 109J:

- Did the authors localise the 109J Bd1705? If their hypotheses are correct, it should be localised at the convex side like the HD100 Bd1705.

- Early in the manuscript, the authors compare bd1705 in HD100 and the non-vibrioid strain 109J, and find a truncation in the N-terminus of the 109J homolog. Interestingly, 109J bd1705 does not complement curvature and muropeptide profiles of Δ bd1705 HD100 when added on a plasmid, unlike the HD100 bd1705. The authors could strengthen their claims on predator entry time, intraperiplasmic curvature and bdelloplast shape by also including these two strains (and the mutant carrying the empty plasmid) in their experiments.

- Line 150: is the bd1705 homolog also expressed in the growth phase of 109J?

- Lines 155-156: when complementing with Bd1075 from HD100 in the WT 109J background, could these proteins form heterodimers and hide the actual function? Complementing a 109J deletion mutant would better support their conclusions on functional differences between the Bd1705 homologs in both strains.

- Did the authors test whether entry time of the non-vibrioid 109J strain is slower than HD100? This could be a nice way to strengthen their claim, if available complementation and cross-complementation strains are also tested.

- The data demonstrating the LD-carboxypeptidase activity of Bd1705 are the strongest part of the manuscript (with the structure), in my opinion. Did the authors also purify the 109J homolog, which is expected to show no LD-CPase activity on purified sacculi?

- Supplementary figure 3c: please indicate the source of the WT 109J DNA sequence reads.

- Supplementary Figure 1b:

- please specify at which cell cycle stage were these RNAseq reads acquired.

- Another RNAseq dataset was published (Karunker et al, 2013): are they consistent with the transcription pattern and the transcription initiation site of bd1075?

- The authors could explain better why they propose a new annotation of the start codon of bd1705 (and not of the 109J homolog, Supplementary figure 3a).
- Since the authors propose a new start codon and SD annotation, showing a few more bases upstream the annotated start would better illustrate the absence of obvious SD in the previous annotation.

- Supplementary figure 7:
 - why is the negative predation control (B. bacteriovorus-free solution after filtration) different from the one in Supplementary figure 6 (heat-killed B. bacteriovorus)?
 - the authors indicate that the predator numbers engaged were lower for the mutant than for the wt. These data should be shown, and used to normalise the biofilm quantification.

- Line 28: "Self-wall curvature" is confusing and "self" is not necessary in my opinion (the sentence already says that the carboxypeptidase activity occurs in the Bdellovibrio strain).

- Line 56: replace antimicrobial by antibacterial

- Line 135: how do authors calculate the "mean curvature"? It would be helpful to indicate how the arbitrary units of mean curvature used in this manuscript relate to more concrete shapes, e.g. what would be the A.U. value expected for a straight line, a circle, a 270° curve, ... This would also help understanding the actual meaning and extent of the curvature differences reported in the paper.

- Line 205: the authors did not really investigate replication, but rather observed predator morphology during cell growth.

- Lines 219-220: "despite becoming more curved by the spherical bdelloplast environment over time" is speculative. The authors cannot exclude the existence of other curvature-mediating factors in B. bacteriovorus during growth, although the confined space seems a plausible cause.

- Figure 6: when fused with mCherry, the complementation (Figure 6d) is stronger than in Figure 1c. How do the authors explain this?

- Lines 719-720: it is unclear where the foci analysis was used in the manuscript, as Bd1705 does not form foci.

We would like to thank all the reviewers for the time they took to review our manuscript and for their helpful suggestions towards improving it. Our responses to their comments are detailed below.

Reviewer #1 (Remarks to the Author):

The paper by Banks et al. reports on a novel mechanism of curved shape generation in *Bdellovibrio*. This is a lovely paper, well-written and easy to follow, with well-controlled and carefully interpreted experiments reporting a significant advance. I only have a few minor suggests/comments.

1. Please supply **movies** underlying the analyses for Fig. 2 as supplementary material (which is currently marked as “Supplementary Information is available at [x]”)

We thank the reviewer for their complimentary response to our manuscript. At the reviewer's request we have now added three new movie files which underlie the analyses for Fig 2: Supplementary Movie 1 (an example of wild-type prey invasion), Supplementary Movie 2 (an example of $\Delta bd1075$ prey invasion), and Supplementary Movie 3 (an example of complemented $\Delta bd1075$ prey invasion). We also include image stills for each of these videos in the new figure addition Supplementary Fig. 8.

2. I am intrigued by their model of how *bd1075* contributes to curvature (particularly the chicken vs. egg problem that arises). I like the idea that mechanical stress might produce some degree of curvature (which is also consistent with their data in Fig. 3B showing a time-dependent increase in curvature even in the mutant), which is then reinforced by PG modifications. Some observations in *E. coli* (PMID 28737752) seem to suggest that this is in principle possible, and it might be useful to incorporate this paper into their discussion.

We agree with the reviewer that the “chicken and egg” nature of curvature generation inside a prey bdelloplast is indeed intriguing. We thank the reviewer for highlighting the above *E. coli* artificially-constrained shape study and we have now incorporated this reference along with two related studies into the manuscript discussion (lines 466-472).

3. Regarding the discussion in line 484 – enhanced L,D CPase activity might actually promote L,D crosslinking (rather than reduce overall crosslinking as stated) by supplying more tripeptide substrate for L,D TPases, or rather induce a shift from DD to L,D crosslinks (and this might in some way promote curvature). Did you look specifically at 3-3 crosslink levels in the mutant by any chance? This effect might be too local to be globally detectable though.

We appreciate this excellent point made by the reviewer. Indeed, all the main muropeptides we identified in Table 1 were either monomers or contained DD-crosslinks. There are approximately 20 LD-TPase enzymes in *Bdellovibrio* (compared to just 6 in *E. coli*) and it would be interesting to ascertain the contribution of these to PG composition in future studies. We have removed the sentence in line 484 and in the revised manuscript, we now state that the reason for the slightly higher overall cross-linkage in cells lacking *bd1075* remains enigmatic (lines 419-421). This would be an interesting point for further study but is beyond the scope of this manuscript.

Reviewer #2 (Remarks to the Author):

This manuscript by Banks et al. presents a study on the role of the protein Bd1075 has on determining the morphology of the vibrioid predatory bacterium *Bdellovibrio bacteriovorus*. Sequence alignment studies suggests that the hypothetical protein is comprised of two domains, an N-terminal L,D-carboxypeptidase (LDCP) domain and a C-terminal NTF2 domain of unknown function. Experiments are described that demonstrate the activity of Bd1075 as an LD-carboxypeptidase, its crystal structure, and the role of the NTF2 domain in localizing the enzyme to the outer curved face of *B. bacteriovorus*. The authors conclude that Bd1075 is indeed responsible for curvature of the bacterium and that it contributes to the overall efficiency and fitness of this bacterial predator.

The study is novel and provides important insight regarding structural determinants for bacterial cell morphology. The manuscript was prepared with great care and attention to detail while being concise; it is clear and easy to read. This reviewer has only a few relatively minor issues that the authors should address to enhance the manuscript.

1. Lines 92-110: This text of the introduction re-iterates much of the Abstract as it reports on the findings of the current work. As such, it could be abbreviated significantly.

Firstly, we would like to thank the reviewer for their kind comments on the novelty and detail in our manuscript. Regarding the end of the introduction, we agree with the reviewer that some of this language could seem repetitive. We have therefore shortened the section referred to by the reviewer to make this more concise. This now forms (the abbreviated) lines 74-85 in the revised manuscript.

2. Line 157: The results presented in the text above suggest that Bd1075 contributes to the curvature morphology but not necessarily the (only) determinant. For example, it cannot be discounted at this juncture that Bd1075 works in concert with one or more other proteins/enzymes for this curvature.

The text at line 127 has been altered to reflect this possibility.

3. Line 300: The authors need to provide the evidence for the purification of recombinant Bd1075 to near homogeneity. Perhaps this could be accomplished with the presentation of SDS PAGE analysis together with the SEC result presented in Supplemental Fig. 11 (i.e., add the SDS PAGE analysis now making it Supplemental Fig. 9.

We have now added Supplementary Fig. 12 showing purification of Bd1075 on an SDS-PAGE gel and updated Supplementary Fig. 15 to show the corresponding gel filtration trace for this purification. Bd1075, as used in our study, is essentially pure.

4. Legend to Figure 4a-d: The authors need to make it clear to the uninformed reader that the “reduced muropeptides” were “released” from digested samples of isolated peptidoglycan; as it reads now, the reader might think the muropeptides had been naturally released.

We thank the reviewer for this point and have amended and clarified the legend to Fig. 4.

5. Legend to Figure 4f: Were these structures confirmed by MS analyses or simply assigned based on elution times and comparison to published studies; the Methods are silent on the use of MS? This should be made clear here and in the legends to Supplemental Figs 8–10 as well.

We thank the reviewer for this point and apologize for having not provided this information in the original manuscript. We assigned the major mucopeptides (no. 1-7) by comparing their elution times to those of the corresponding, well-characterized mucopeptides from *E. coli* (references provided in Methods). We collected the mucopeptide fractions no. 8-19 and analysed these by mass spectrometry. We have now added a table with the measured and theoretical masses of these mucopeptides (Supplementary Table 2), added further information to the methods (lines 592-598 and lines 609-610) and also added this to each figure legend as requested.

6. Table 1: The table legend states that the values represent the “relative percentage area” of each mucopeptide peak in Fig. 4. The Methods section for this experiment simply refers to Reference 48, but were the appropriate response factors for the various mucopeptide oligomers used for these quantifications? If so, this should be made more clear here and stated in the Methods as the data are reporting more than simple area under the curves for each identified HPLC fraction. If not, and simple areas are indeed reported, then they should be manipulated by the appropriate response factors to provide a more accurate comparison of the mucopeptide fractions. This applies here and also to Supplemental Table 1.

We appreciate the thoughts of the reviewer regarding the quantification of mucopeptides. However, we respectfully disagree with the notion that converting the peak areas to molar percentage by using the theoretical conversion factors from (Glauner, 1988) will provide a “more accurate comparison of mucopeptide fractions”. The quantification by peak areas is a valid approach in analytical chemistry that has been used in numerous studies in the literature (including for mucopeptide composition). Most importantly, calculating with the conversion factors would actually reduce the accuracy. This is because the conversion factors are theoretical values estimated from the number of disaccharide units, amide bonds, 1,6-anhydro groups and Lys-Arg residues (which are not present in our samples) in each mucopeptide, calculated with the following formula:

$$C = D / (0.6 \times D + 0.1 \times A - 0.1 \times E + 0.2 \times L) \text{ (Ref 48).}$$

with C, correction factor; D, number of disaccharide units; A, number of amide bonds; E, number of 1,6-anhydro groups; L, number of Lys-Arg

We notice that each of the multipliers (0.6; 0.1; 0.2) used in the formula has only 1 decimal. The resulting value for C, which is generated as the sum of 1-decimal theoretical values, is not very accurate and has certainly less accuracy than the experimental values determined for the peak areas; thus manipulating the data with the conversion factors would decrease the accuracy of our comparisons. Moreover, the values for C are 0.9, 1.0 or 1.1 for all mucopeptides we detected, except for Di which has a C of 1.3. Considering all aspects we believe that our comparisons are best done with the area percentages, without manipulating the data by the correction factors.

Reviewer #3 (Remarks to the Author):

Review Banks et al

In this manuscript the laboratories of Dr. Sockett and Dr. Lovering study the role of Bd1075 in generating cell shape in the predator bacteria *B. bacteriovorus*. Using a wide collection of methods, their main findings are that Bd1075 is:

- 1 - important for generating vibroid morphology
- 2 – important for optimal predation
- 3 - a LD-Cpase, and that its enzymatic function is important for vibroid morphology
- 4 – localizes asymmetrically via its NFT domain, which is also important for vibroid morphology

This is a nicely constructed, and reported study. I have just a few small issues:

1- The complementation results are a little confusing. Most of their figures show only partial complementation of the vibroid shape by expressing the wildtype allele (Figure 6e is different and it seems like the mCherry fusion complements more?). However, their mass spec data show that the complementing strains may be 'overly complementing', in that the complemented strain has no monomeric tetrapeptides. I think it would be worth discussing this discrepancy somewhere in the discussion.

We thank the reviewer for their kind words and considered reading of our manuscript and for raising this interesting discussion point.

For the biochemical sacculi experiment (Fig. 4), we chose to use plasmid-based complementation as multiple copies of the gene may increase the likelihood of observing Bd1075 biochemical activity *in vivo* which we did obtain. For Fig. 6e Bd1075-mCherry fusion complementation, it was more appropriate to use genomic single-crossover based complementation to represent a more realistic regulatory system and this resulted in strong complementation of cell curvature. However, the question about why the sacculi analysis showed "over-complemented" enzymatic activity of Bd1075 on PG, while the measured bacterial strain curvature (also plasmid-complemented) was only partially complemented (Fig. 1c) is interesting. We note that Sycuro *et al.*, (2013) observed the same phenomenon for LD-CPase homologue Csd6 and this was due to a 2.1X increase in Csd6 protein production compared to the WT. This resulted in over-cutting of PG but only partial shape complementation therefore overabundance of a cell shape protein appears to perturb cell shape. This suggests that "over-cutting" results in a more general dysregulation of the system which results in "under-complementation" of curvature.

We have added this idea to the manuscript discussion to assist the reader in interpreting this complexity (lines 408-417).

2- For the single cell data, the replicates are all shown together, and a p-value is calculated between the averages of the strains (a very high N, giving a very small p-value). However, it is more rigorous to separate out the replicate data (this can be done as a 'super-plot') and calculate a p-value comparing the average of the averages (so an effective N of 3).

The reviewer raises a good point and this can be a good way to analyse certain data. However, for our bacterial cell experiments, there is inevitable day-to-day biological variation (partly caused by time, oxygen and temperature variations during long time periods when samples are being viewed on the microscope), and therefore we believe that acquiring data for as many cells as possible and grouping these together for analysis for the main figures is the best way to analyse our datasets and to reveal and present phenotypes fairly. We believe our data very strongly support our conclusions of curvature of bacteria.

To address reviewers' concerns, we have made amendments to improve our presentation of results: we have updated Fig. 1c and Fig. 6e to instead show the median curvature values rather than the mean since the data were non-parametric. We also include full data distribution plots for Fig 6b,d,e,f and g in Supplementary Fig. 10.

Reviewer #4 (Remarks to the Author):

Banks et al. report that the Bd1075 protein is involved in setting *Bdellovibrio bacteriovorus* cell curvature. The authors state that the vibrioid morphology increases the evolutionary fitness of predatory bacteria, improving prey invasion and intracellular growth. They investigate the protein domains responsible for its intriguing localisation to the convex side of the cell. The authors also demonstrate that Bd1705 is an LD-carboxypeptidase and solved the structure of the Bd1705, revealing interesting features compared to other related PG-modifying enzymes. Overall, although potentially important not only for the *Bdellovibrio* research field but also for increasing our knowledge of bacterial shape determinants and their physiological roles, this study raises several concerns. While the data clearly indicate a role of Bd1075 in cell curvature, interesting structural features and its LD-carboxypeptidase activity, several major conclusions are not fully supported by data in the present state and, in my opinion, require additional controls and stronger statistical analysis.

We thank the reviewer for their appraisal of our manuscript and address their concerns below.

Major comments

I have 3 major concerns about this manuscript.

1) My biggest concern relates to the way most data from in vivo experiments are analysed.

- At several instances, the mean is inappropriately used as a statistical value to compare populations that show non-normal distributions (methods indicate that normality was verified but this result is very surprising considering data shown in Supplementary Figure 5). Besides, means are sometimes calculated from several replicates, but the distribution of all data points indicates quite some variability between repeats (Supplementary Figure 7), questioning the relevance of the mean as a proxy for statistical comparison of conditions/strains. Along the same line, some figures (e.g. Figure 2b-c, Figure 4b-c) presenting means calculated from several replicates would benefit from showing all data points or full distributions and their source (i.e. which replicate) instead of the single mean value only.

We tested data for normality by the D'Agostino-Pearson test in Graphpad Prism and determined that distributions were non-normal (as stated in methods section 'statistical analysis'). Regarding the presentation of the mean \pm standard error in Fig. 1c and Fig. 6e, we thank the reviewer for raising this issue. The reviewer is right - since the data are non-parametric, it is more appropriate to present medians and we have now corrected these graphs to show the median \pm 95% confidence intervals. We also correct measurements stated in the text to reflect this (lines 124-126, 150-154, 191-200). Thank you for raising this point.

For Fig. 3 bdelloplast morphology graphs, however, presentation of the median rather than the mean makes it more difficult for the reader to understand and appreciate the microscopy

phenotype. There are statistically significant differences between the wild-type and $\Delta bd1075$, therefore we continue to present the mean for this figure. However to address the reviewer's points, we now also include a graph showing the median \pm 95% confidence intervals in a new supplementary figure (Supplementary Fig. 9). In addition, we now include floating bar plots to show the full spread of the data for Fig. 3 in Supplementary Fig. 10. For Fig. 2b-c (attachment and entry times), the data are currently presented as box and whisker plots which already show the full spread of the data with medians and interquartile range. The means were not used for statistical comparisons (please see below).

Importantly, these changes to data presentation do not affect the results from statistical tests nor the conclusions that we draw in this study.

- It is unclear how the authors chose particular statistical tests to compare means in different figures (e.g. Kruskal-Wallis test for Figure 1c, Mann-Whitney test in other figures), and whether the chosen tests are the most appropriate in cases of non-normal distributions.

All data were tested for normality by the D'Agostino-Pearson test in Graphpad Prism. All data distributions were non-normal, therefore we used appropriate non-parametric tests: Mann-Whitney for comparison of two groups and Kruskal-Wallis for comparison of multiple groups. These tests do not compare means but medians since the data are non-parametric. For all statistical analysis, the statistical test that was applied to the data set is referred to in the relevant figure legend.

- The authors should clarify or modify their analysis of the prey-death curves in Supplementary figure 6, which aims at comparing the predation efficiency of the $\Delta bd1705$ mutant with the wild-type strain. Regarding the negative predation control in panel a: heat-killed predators were added, so no prey death is expected. Could the authors explain why they see a decrease of luminescence in a, and whether the control values were taken into account in panel b? Why using the area under the curve as a comparison parameter, especially considering that the test curves are overlapping with the negative control curve for half of the time? Calculating the slope of each prey-death curve would seem more straightforward in my opinion. Is the luminescence increase until 4h due to prey growth/proliferation? It is unclear what each data point represents in panel b and how this type of calculation supports the absence of difference between the two strains. Points seem quite spread around the linear fit. Is fitting not affected by the precision limit of plaque counts?

These are interesting questions which we address below, but as we state in the manuscript, there is no difference between the wild type and $\Delta bd1075$ by this assay.

The luminescence assay monitors an oxygen-requiring production of luminescence by *E. coli* prey which occurs in live metabolising bacteria until they are killed by multiple early predation events. Only one of these events is the actual invasion process we measured for wild-type and $\Delta bd1075$ by time-lapse microscopy. Prey killing abolishes prey luminescence.

The conditions in which best to observe this have been carefully optimised previously (Lambert et al., 2003). Using buffer only, when this assay was developed and published, did not control well for the extra nutrients provided to the *E. coli* prey by the addition of the *Bdellovibrio* (mixture of media carryover and some dead predators) and therefore heat-killed cells were used as a control. The initial increase in luminescence is a result of initial growth of the prey cells and availability of nutrients. The depletion of nutrients and the diminishment of some oxygen by metabolism of the prey over the course of the 24 h experiment led to a decline in luminescence in the control, but despite this, the reduction in luminescence resulting from *Bdellovibrio* killing could easily (and confidently) be measured.

The reviewer is correct that precision of plaque counts is limited with significant natural variation within a complex biological interaction (that depends on metabolism of two bacterial types over 24 h) in this real assay of live predation. Five independent experiments, resulting in many datapoints, were carried out to overcome these inherent limitations. Therefore, we chose to compare area under the luminescence curve relative to plaque counts to incorporate all of these factors (panel b) without the need for correcting for negative control as this was the same for both strains. In this way, the points in panel b represent the rate of drop in luminescence relative to initial numbers of predators added as determined by plaque count, and show no difference between the strains.

To address the reviewer's concern, we have additionally analysed the drop in luminescence curves by correlation analyses for each dilution series. In every case, comparing wild-type to mutant resulted in a significant Spearman's correlation (non-parametric test), with $p < 0.0001$, further confirming that there is no significant difference between the strains. This has been added to the legend of Supplementary Fig. 6.

- Data in Figure 3 do not convincingly support the conclusion about vibrioid shape being a fitness advantage for *B. bacteriovorus*. The frequency of the clear cases of bdelloplast elongation (panel c) should be calculated and compared between wt and $\Delta 1705$ strains. The authors only refer to a "sub-set" at Line 221. Could this sub-set be responsible for the changes in mean bdelloplast dimension shown in panels d-g (mean values being sensitive to low numbers of outliers)?

We have now calculated the percentage of cases of bdelloplast stretching by $\Delta bd1075$ by using a cut-off circularity value of 0.96. We find that 0% of WT bdelloplasts are < 0.96 in contrast to 9.2% of $\Delta bd1075$ bdelloplasts which are < 0.96 . We now reference this 9.2% being the sub-set of bdelloplasts which deformed by $\Delta bd1075$ in the manuscript at line 185 and describe this calculation in the methods (lines 659-663).

The authors could strengthen their analysis by testing if the 109J strain significantly deforms the bdelloplast from within.

Since 109J is a completely different isolate of *Bdellovibrio*, it contains multiple genomic differences to HD100 besides absence of cell curvature and also has a longer cell body. Therefore, there are likely to be a great number of differences that could affect this experiment, making it unsuitable to compare these strains directly here.

I am also puzzled by the apparent absence of difference in width (panel g), although bdelloplasts seem narrow in panel c, and related to this, by the apparent increase in bdelloplast area (panel d). How could the area of the bdelloplast change over time? I would expect this parameter be fixed by the initial dimensions of the prey.

Whilst one bdelloplast out of six in panel g appears narrow, our analysis of hundreds of bdelloplasts shows no overall difference in WT vs $\Delta bd1075$ width. The dimensions of the bdelloplast are not necessarily fixed to that of the uninvaded prey cell since many peptidoglycan-modifying enzymes act to modify the cell wall including DD-endopeptidases (Lerner et al., 2012), deacetylases (Lambert, Lerner et al., 2016), LD-transpeptidases (Kuru, Lambert et al. 2017) and other as yet uncharacterised enzymes. The DD-endopeptidases cut cross-links in the prey wall, making it weakened and more malleable with potential for stretching and deformation. Because the diversity of peptidoglycan chemistry across the *E. coli* PG sacculus is still not fully known to science and is especially understudied in the predatory cycle where it is being influenced by many predatory enzymes, we cannot accurately account for the resulting bdelloplast shape deformations that may be due to predatory enzymes targeting different PG chemistries.

I am also not convinced that the statistical significance obtained in panel e (mean \pm SD circularity values: 0.98 ± 0.04 vs 0.99 ± 0.01) relates to a physiological difference in shape.

The software MicrobeJ (Ducret et al., 2016) was used for all measurements of morphology. Whilst the differences in circularity appear small, this is a result of how this is calculated in the software. This defines circularity as $4\pi \times \text{area} / \text{perimeter}^2$ with a value of 1.0 indicating a perfect circle. To make it clearer that these seemingly small differences are significant, we have added this explanation to the appropriate methods section (lines 659-660) and separate the methods text into two headings: 'phase-contrast and epifluorescence microscopy' and 'image analysis' in order to add more detail to the latter section.

We also attach a reviewers' figure showing a gallery of example cells annotated with their circularity and also curvature values obtained in MicrobeJ. This also addresses the reviewer's request for an explanation of curvature values further below. Most circularity values are close to 1.0 – even for rod shapes (often >0.8 A.U.) therefore a seemingly very small difference in circularity A.U. values e.g. 0.98 vs 0.99 does translate to a meaningful difference in cell shape.

2) My second concern relates to the interpretation of the sub-cellular localisation data.

- Supplementary figure 12. It is confusing that the Bd0064-mCerulean signal is thinner and shorter than the Bd1705-mCherry signal in the corresponding cells (where the opposite is expected). This might come from (too) strong contrasting of the Bd0064 images (and additional processing mentioned in the methods). I suggest that the authors refine their image display.

Bd1075-mCherry fluorescence is naturally much dimmer than that of the tagged highly abundant cytoplasmic control protein Bd0064 (a common phenomenon for *Bdellovibrio* proteins) and therefore it is necessary to adjust the brightness/contrast in the different fluorescent channels to clearly visualise the location of the proteins. The small size of *Bdellovibrio* cells are approaching the resolution limitations of light microscopy. However, in response to the reviewer, we have updated Supplementary Fig. 16, having toned down the brightness of the Bd1075-mCherry signal. We also note that other than adjustments to brightness/contrast and sharpening/smoothing the overall image, no additional image manipulation has been carried out on any images.

- The authors claim at several places that the predicted N-terminal signal sequence targets Bd1705 to the periplasm. Periplasmic localisation of the protein is however not shown. This is an important missing piece of data considering that the authors conclude on the cell wall modification role of this protein.

Since Bd1075 contains an N-terminal sec signal sequence and the protein acts on peptidoglycan in the periplasm, the evidence strongly suggests a role for Bd1075 in the periplasm. To answer this more directly, we have now constructed two new strains and performed an additional experiment. Unlike mCherry, the fluorophore mCitrine does not fluoresce in the periplasm (most fluorophores cannot). We made a fusion of Bd1075 to mCitrine, verified by sequencing that it is fully correct, and show that this does not fluoresce, whereas a control of cytoplasmic protein Bd0064-mCitrine does fluoresce, providing further evidence that Bd1075 localizes to the periplasm. These new data form Supplementary Fig. 17 and we have added this to the manuscript results (line 318 and lines 323-327) and methods (lines 563-565 and line 634).

- I have several concerns related to the interpretation of data in figure 6c-d, where Bd1705 variants are produced in fusion with mCherry, either in presence of the wild-type untagged Bd1705 copy or in a bd1705 null strain.

(i) The authors do not fully consider the localisation patterns of the variants in the $\Delta bd1705$ strain (shown in d) to conclude on the impact of certain residues and domains on the localisation of Bd1705. Yet it seems that the wild-type copy impacts their fusions, possibly in different manners than suggested. For instance, they claim that the C156 of the catalytic domain is not required for localisation (lines 413-415), but the same mutant is unable to localise at the convex side in the absence of the wild-type Bd1705, suggesting that C156 does have an impact on the subcellular localisation, and that it localises at the convex side only if the wild-type protein is there.

While the authors propose that this is due to curvature sensing by the C156A mutated form (lines 580-585), it is also possible that it is simply recruited by the wild-type form. They have not excluded the possibility of self-interactions in *B. bacteriovorus* in vivo and therefore the claims from this figure should be modified.

As discussed earlier, there is certainly an interesting, multi-part, “chicken and egg” process in *Bdellovibrio* curvature generation. We believe that our fluorescence localization data in Fig. 6 convincingly demonstrate that the NTF2 domain of Bd1075 targets the protein to the convex cell face. However, we agree with the reviewer (who discusses this further later) that the possibility of Bd1075 self-recruitment to the convex face is an idea that should not be excluded. We therefore rephrase the original lines 580-585 to include the reviewer’s suggestion of Bd1075 self-interactions, forming the new lines 516-521 in the revised manuscript.

Regarding Fig. 6, we offer further explanations below to clarify experimental details to the reviewer:

The LD-CPase active site mutant protein C156A can localize correctly to a convex face, but **not** generate final full curvature. In Fig. 6c, that final full convex face is generated by the presence of a wild-type copy of *bd1075* in the strain background. When C156A-mCherry is introduced into this strain, the fusion localizes to the convex face. Therefore C156A does not negatively perturb Bd1075 localization because wild-type Bd1075 has generated a final full convex face, as found in WT strain HD100.

We go on to provide further evidence in Fig. 6d which uses a $\Delta bd1075$ recipient background (rod-shaped cells with no wild-type copy of *bd1075* in this instance). When C156A-mCherry is introduced into this recipient, there is no final convex face for the fusion protein to now localize to. As C156A is a catalytic LD-CPase point mutant, it cannot generate curvature itself. Thus, although the protein could localize correctly (demonstrated in Fig. 6c), it cannot generate cell curvature and therefore there is no convex face at which to observe the mCherry fusion.

The most important finding from Fig. 6, however, is that the C-terminal NTF2 domain of Bd1075 targets the protein to the convex face. We demonstrate this directly by showing that Bd1075-mCherry targeting is abrogated by either an NTF2 truncation or point mutation (Fig. 6c) and that neither of these mutant proteins can complement $\Delta bd1075$ curvature (Fig. 6d).

(ii) The heat map of Y274A in panel c seems to indicate an intermediate pattern (between the E302 mutant and WT), which is not commented (the authors conclude on the same inability to localise at convex sites for both E302 and Y274A mutants, which is the case in the $\Delta bd1705$ but not in the wt background). The conclusions drawn from these data (i.e. that the C-terminal NTF2 extension is important for localisation) should be reconsidered: the NTF2 domain is important for localisation but a residue in the catalytic domain as well.

We appreciate the reviewer noting a slightly intermediate localization pattern for the Y274-mCherry expressed in $\Delta bd1075$. We have thus re-phrased the text (lines 361-364) to suggest that this may be because the mutation might partially but not completely destabilize

the NTF2 pocket, hence a proportion of protein may be able to localize. As discussed above, our data show that the LD-CPase domain is not required for correct localization to an outer convex curve (Fig. 6) - only the NTF2 domain.

(iii) It is crucial to confirm by western blot that all fusions in all backgrounds are properly produced and not truncated (which would affect the distribution of the fluorescent signal).

At the reviewer's suggestion, we have now carried out western blot analysis using an anti-mCherry antibody for all Bd1075-mCherry fusions (5 fusions in the wild-type background and 5 fusions in the $\Delta bd1075$ background) with appropriate controls. We can see correct production of each Bd1075-mCherry fusion protein at the expected sizes, indicating that each protein is produced correctly. This forms a new figure: Supplementary Fig. 18. We now include this control in the manuscript text (lines 339-340 and lines 375-376) and have added the experimental methodology to the supplementary methods (lines 131-143 in the Supplementary information).

(iv) The authors state that the C156A-mutated fusion that do not display asymmetric localisation remains localised to the cytoplasm. The fluorescent signal seems indeed to fill the whole cell. How would this mutation impact the periplasmic targeting of the protein?

The C156A mutated fusion protein has a complete sec signal. The phrase used was "remained localized at the center of the rod-shaped cell", which we agree is confusing and we have changed to "was diffuse throughout the rod-shaped cell" (lines 379-380). The localization of C156A and targeting to the convex cell face is discussed in the points above.

3) Several main claims are not supported by data and should be toned down.

- Based on data in Figure 2, the authors draw the important conclusion that entry time is longer for the mutant than for the wild-type. (i) Complementation is needed to support the claim, especially since the strain is available. (ii) The time difference being very small (4.3 ± 0.9 min vs 6.1 ± 1.7 min) and using means (despite an apparent non-normal distribution for $\Delta bd1705$), I am not fully convinced that it is physiologically relevant (how does it compare with the time-resolution of the experiment / with the variability of this event?). (iii) The authors then go further in their conclusion by saying that these data, combined with the curvature differences observed in Figure 1, indicate that the vibrioid morphology facilitates the traversal of predators across the prey cell envelope into the periplasmic compartment of the rounded prey cell. This claim should be toned down, since they have not shown directly that the curvature defect in cells lacking *bd1705* is what causes the reported entry delay (in addition to the apparent weakness of these data in their present form).

To address the reviewer's concern here, we have now carried out the suggested experiments of attachment and entry time analysis (with exactly the same parameters as before and 3 repeats) using a strain which complements $\Delta bd1075$. The complemented strain enters prey significantly faster than the $\Delta bd1075$ mutant, and it shows no significant difference in prey entry speed from the wild-type. This has been added to the manuscript (lines 157-159) and Fig. 2.

We did not use means as the data was non-parametric – Fig. 2 shows the full spread of the data as a box-and-whisker plot with the central line indicating the median. The appropriate non-parametric statistical test was used (Kruskal-Wallis) to compare the three strains. We believe that this strengthens Fig. 2 (which has now been updated to include the new data). As stated earlier, we have amended lines in the text to describe the measurements using median values (4.0 min for WT and 6.0 min for $\Delta bd1075$). Although the difference in entry time is on the scale of minutes, this translates to a physiologically relevant difference since

rod-shaped *Δbd1075* predators are entering prey 50% more slowly than curved wild-type cells. Curvature could therefore confer an important advantage and competitive edge to curved *Bdellovibrio* predators which can enter and occupy their prey relatively more quickly.

- The claim that the vibrioid morphology confers optimal intracellular growth (e.g. line 468) is overstated. The data do not show that the modified bdelloplast shape and the lower curvature of the predator cells impact their growth per se. For this, analysis of growth rates or number of predator cells released per bdelloplast would be needed. The absence of obvious predation efficiency difference between the WT and the *Δbd1705* cells suggests instead that there is no major benefit for growth conferred by curvature. This claim about optimal growth / evolutionary advantage provided by curvature should be softened.

We acknowledge and agree with the reviewer's point and have thus softened the language associated with growth inside bdelloplasts at a number of points throughout the manuscript: the abstract (lines 30-32), introduction (lines 83-85), results (lines 202-203) and discussion (line 523).

- The claim that the C-terminal NTF2 domain targets the protein is not fully supported by the data, showing that a mutation in the catalytic domain also prevents the localisation of the Bd1705 protein fusion to the convex lateral side (see major comment #2).

This point has been addressed above.

- At several places in the manuscript, the authors write "what becomes" the convex cell face (e.g. lines 385, 410, 434, 442, 542). However, at this point, the authors have not demonstrated that the proposed asymmetric localisation of Bd1705 arises before the establishment of curvature. This would require inducing the expression of a Bd1705 fluorescent fusion over time in a *bd1705*-null background (straight cells) and the observation of curvature increase after asymmetric positioning of Bd1705. Using "what becomes the convex face" might also sound counter-intuitive, since the mother *B. bacteriovorus* cell is already curved before it starts to grow. Please clarify or modify throughout.

We agree with the reviewer that this terminology may be a little confusing due to the "chicken and egg" nature of *Bdellovibrio* curvature creation. We have removed mentions of "what becomes" from the manuscript. Induction of *bd1075* in a *Δbd1075* mutant would be interesting but we are unfortunately unable to perform this due to a paucity of certain genetic tools for *Bdellovibrio* which includes genetic induction (neither arabinose nor IPTG induction work). Moreover, *Bdellovibrio* grows as a filament before cell division so stable plasmid replication and retention is challenging.

Minor comments

- Discussion, lines 531-537 / 585-590: The idea of topological sensing the physical space to prime asymmetric curvature (and therefore curvature-mediated asymmetric protein localisation) is interesting. However, the idea of topological sensing of the physical space and its impact of bacterial morphology is not a new concept, unlike what the text seems to suggest (see for instance Männik et al, PNAS 2012; Wu et al, Nat Nanotech 2015; Söderström et al, Nat Commun 2018). Curvature-mediated localisation has also been described for many proteins in various species.

We agree with the reviewer that topological sensing of physical spaces by bacteria is an interesting concept. We did not intend to imply that this is the first study to suggest the idea but can see how it could have been thus interpreted. To address this we have rephrased lines 518-521 at the end of the discussion accordingly. Thank you for highlighting those studies. We have included a new paragraph to discuss topological sensing and adaptation to

different environmental shapes (lines 466-472) which includes references to the above papers.

Besides, I am also thinking of alternative explanations for the asymmetric localisation of Bd1705. In fact, it is not clear why Bd1705 would help fixing a bdelloplast-induced curve. Is there evidence that the growing *B. bacteriovorus* always “sticks” to the bdelloplast membrane? I would assume that the filament is sometimes twisting in the middle of the bdelloplast, where curvature cannot be imposed by the bdelloplast boundaries.

Bdellovibrio are often seen to curve around the outer wall of the bdelloplast and while true that they are capable of twisting through the middle of the bdelloplast as suggested, 3D-SIM images show that they often grow spiralling along the perimeter of the bdelloplast (Kuru, Lambert et al., 2017). It is likely that the turgor pressure of the bdelloplast interior could push the *Bdellovibrio* to the outer membrane and wall of the bdelloplast but that would be the subject of further studies.

How about the curvature of free-living *B. bacteriovorus* variants? If these are curved, is their shape modified in the absence of bd1705?

If yes, that would suggest that there is no need for a bdelloplast-induced curve at the first place, and that Bd1705 induces curvature (directly or not) independently of the physical constraints of a spherical and confined environment (“topological sensing”). A perhaps more straightforward interpretation of the asymmetric localisation of Bd1705 would simply arise from the presence of Bd1705 in the mother cell, which could recruit and direct the localisation of newly synthesised Bd1705 on that side in the growing *B. bacteriovorus*. This alternative hypothesis would be in line with the idea that self-interaction and recruitment, rather than topological sensing, might determine asymmetric localization of Bd1705 in *B. bacteriovorus*.

This paper is about predatory growth not host-independent (HI) *Bdellovibrio* growth which is a very poorly characterised biological system. HI non-predatory *B. bacteriovorus* strains are highly pleomorphic with different asynchronous life cycle stages and likely have dysregulated gene expression vs predatory cells. Therefore we did not interrogate that biological system here.

As mentioned earlier, we appreciate the reviewer’s suggestion of an alternative model in which Bd1075 may recruit itself to the outer curve (“chicken-and-egg”-like) and we have incorporated this possibility into our discussion (lines 516-518).

- The authors write that most known cell shape determinants are localised at the inner concave face of bacteria. Are there many more of those beside crescentin and CrvA/B? If not, then the convex localisation of Bd1705 and CcmA is just as frequent (2 cases concave vs 2 cases convex).

We thank the reviewer for raising this. Aside from MreB which localizes to regions of negative Gaussian curvature like concave faces, we don’t think there are further examples of concave-localizing shape proteins besides crescentin, CrvA and CrvB. We have therefore removed the sentence in the results stating this (lines 393-396 in the original manuscript) and rephrased line 478 and lines 482-484 in the discussion to address this. We have also removed “in contrast to most known shape-determinants” from the abstract.

- The order of appearance of supplementary figure references in the text is generally not following the order of these supplementary figures. Please reorder supplementary figures and/or panels.

We have re-checked the ordering of supplementary figure references.

- 109J:

- Did the authors localise the 109J Bd1705? If their hypotheses are correct, it should be localised at the convex side like the HD100 Bd1705.

Our data suggest that the 57 amino acid deletion in *bd1075*_{109J} causes the protein to become completely dysfunctional (likely misfolding and being targeted for degradation) therefore we did not pursue this.

- Early in the manuscript, the authors compare *bd1705* in HD100 and the non-vibrioid strain 109J, and find a truncation in the N-terminus of the 109J homolog. Interestingly, 109J *bd1705* does not complement curvature and mucopeptide profiles of Δ *bd1705* HD100 when added on a plasmid, unlike the HD100 *bd1705*. The authors could strengthen their claims on predator entry time, intraperiplasmic curvature and bdelloplast shape by also including these two strains (and the mutant carrying the empty plasmid) in their experiments.

As the reviewer recommended, we strengthened our data showing a difference in prey entry time for curved wild-type HD100 vs Δ *bd1075* by complementing Δ *bd1075* and demonstrating that this strain returned to a wild-type entry time (see above and revised Fig. 2). We therefore didn't think it necessary to assess this any further.

- Line 150: is the *bd1705* homolog also expressed in the growth phase of 109J?

RT-PCR shows that the *bd1075* homologue in WT 109J is expressed in attack-phase (Supplementary Fig. 3d). In WT HD100, *bd1075* is constitutively expressed. As 109J shares complete homology with the *bd1075* promoter region of HD100, we consider it highly unlikely that the 109J version is regulated differently. We also show in the manuscript that this homologue is almost certainly non-functional due to its N-terminal truncation.

- Lines 155-156: when complementing with *Bd1075* from HD100 in the WT 109J background, could these proteins form heterodimers and hide the actual function? Complementing a 109J deletion mutant would better support their conclusions on functional differences between the *Bd1705* homologs in both strains.

Expressing *bd1075*_{HD100} in WT 109J (which is naturally rod-shaped) increases its cell curvature and *Bd1075*_{HD100} is catalytically active on WT 109J PG (Supplementary Fig. 11c and 13b). As the *Bd1075*_{109J} homologue is inactive (Supplementary Fig. 11a), complementing a deletion strain would almost certainly have the same effect.

- Did the authors test whether entry time of the non-vibrioid 109J strain is slower than HD100? This could be a nice way to strengthen their claim, if available complementation and cross-complementation strains are also tested.

As discussed above, we have strengthened the entry phenotype through including a complemented Δ *bd1075* strain. Since 109J is a different strain to HD100 and contains other genomic differences as well as a longer cell length, we did not think it an appropriate comparison to examine 109J entry time.

- The data demonstrating the LD-carboxypeptidase activity of *Bd1705* are the strongest part of the manuscript (with the structure), in my opinion. Did the authors also purify the 109J homolog, which is expected to show no LD-CPase activity on purified sacculi?

Bd1075_{109J} does not have any LD-CPase activity on PG (Supplementary Fig. 11a) so is likely (due to the deletion of a region of the gene) to be misfolded and degraded. Therefore purification would be unlikely to reveal any biologically relevant information.

- Supplementary figure 3c: please indicate the source of the WT 109J DNA sequence reads.

We have now added the source to the figure legend.

- Supplementary Figure 1b:
- please specify at which cell cycle stage were these RNAseq reads acquired.

We have updated the figure legend to specify this.

- Another RNAseq dataset was published (Karunker et al, 2013): are they consistent with the transcription pattern and the transcription initiation site of bd1075?

The Karunker *et al.*, (2013) data set shows transcription in attack-phase (AP) and at 3 h only. *bd1075* is expressed at both timepoints in agreement with our data. Transcription appears to start in a similar place to that described by our data. Our RT-PCR which was performed across many more timepoints (AP, 15 min, 30 min, 45 min, 1 h, 2 h, 3, h and 4 h – Supplementary Fig. 2) is more comprehensive as it shows that *bd1075* is constitutively transcribed across the predatory lifecycle.

- The authors could explain better why they propose a new annotation of the start codon of bd1705 (and not of the 109J homolog, Supplementary figure 3a).

Since the *bd1075*_{109J} gene has 100% identity to that of *bd1075*_{HD100} (except for the 57 residue truncation), it is likely that *bd1075*_{109J} also has an incorrectly annotated start codon and begins at the same start codon we propose for *bd1075*_{HD100}.

- Since the authors propose a new start codon and SD annotation, showing a few more bases upstream the annotated start would better illustrate the absence of obvious SD in the previous annotation.

We have updated Supplementary Fig. 1b accordingly.

- Supplementary figure 7:
- why is the negative predation control (B. bacteriovorus-free solution after filtration) different from the one in Supplementary figure 6 (heat-killed B. bacteriovorus)?

In the liquid predation experiment (Supplementary Fig. 6), heat-killed *Bdellovibrio* are used as a negative no-predator control as this balances the addition of nutrients with the added live *Bdellovibrio* (discussed in detail above). The study by Lambert et al., (2003) demonstrate this necessity. However, for Supplementary Fig. 6, this is not required and therefore a more standard 0.22 µm filtrate is used as a no-predator negative control.

- the authors indicate that the predator numbers engaged were lower for the mutant than for the wt. These data should be shown, and used to normalise the biofilm quantification.

We now present the plaque numbers in Supplementary Fig. 7b. We matched the numbers of *Bdellovibrio* at the start of the experiment as accurately as possible and used plaque counts to confirm that the *Bdellovibrio* were viable in expected numbers. It would not be appropriate to normalise the OD measurements of remaining biofilm after a 24 h complex predation assay directly to these plaque counts (which are notoriously variable, themselves deriving from days of predation in different conditions of soft agar on a lawn of prey). The data

strongly suggest that there is no significant biological difference between the ability of wild-type and $\Delta bd1075$ predators to prey on biofilms.

- Line 28: “Self-wall curvature” is confusing and “self” is not necessary in my opinion (the sentence already says that the carboxypeptidase activity occurs in the *Bdellovibrio* strain).

‘Self-wall’ has now been removed.

- Line 56: replace antimicrobial by antibacterial

This has now been replaced.

- Line 135: how do authors calculate the “mean curvature”? It would be helpful to indicate how the arbitrary units of mean curvature used in this manuscript relate to more concrete shapes, e.g. what would be the A.U. value expected for a straight line, a circle, a 270° curve, ... This would also help understanding the actual meaning and extent of the curvature differences reported in the paper.

Curvature is defined in MicrobeJ software as the reciprocal of the radius of curvature measured between the end points and the center of the medial axis of the cell. We have now added this explanation to the appropriate methods section (lines 644-647). It is not possible to simply measure a straight line or curve alone as the software is designed to detect whole bacterial shapes, however, as stated earlier, we now include a reviewers’ figure illustrating examples of curvature (and circularity) values for different *E. coli* and *Bdellovibrio* shapes.

- Line 205: the authors did not really investigate replication, but rather observed predator morphology during cell growth.

This is a good point and we have re-phrased instances of ‘replication’ with ‘growth’ and have also replaced ‘optimizing replication’ with ‘optimizing growth’ in the manuscript title.

- Lines 219-220: “despite becoming more curved by the spherical bdelloplast environment over time” is speculative. The authors cannot exclude the existence of other curvature-mediating factors in *B. bacteriovorus* during growth, although the confined space seems a plausible cause.

We agree that it is slightly speculative, however, as the reviewer states, the confined space suggests that it is the spherical environment rather than other factors.

- Figure 6: when fused with mCherry, the complementation (Figure 6d) is stronger than in Figure 1c. How do the authors explain this?

Please see the first response to reviewer 3 in answer to this question.

- Lines 719-720: it is unclear where the foci analysis was used in the manuscript, as *Bd1705* does not form foci.

This is a feature of the MicrobeJ software which may be confusing. “Foci” analysis is the most appropriate analysis within MicrobeJ software to detect fluorescence and generate heat maps and doesn’t necessarily refer to specific foci, but is rather a feature of how the software detects fluorescence. This is now explained with greater clarity in the methods (lines 647-652).

Reviewers' Comments:

Reviewer #2:

Remarks to the Author:

Each of my issues have been addressed appropriately/adequately.

Reviewer #3:

Remarks to the Author:

The reviewers have addressed my concerns and I am highly supportive of publishing this very interesting and well constructed study.

Reviewer #4:

Remarks to the Author:

I would like to thank the authors for their time and efforts, and for their response to my questions and concerns including new controls and analysis. I have uploaded the rebuttal pdf, in which I have added comments for each of their response (in green). I have two concerns that I could not include fully in the rebuttal pdf file (lack of space), which I am therefore pasting below (I am referring to them in the corresponding response in the file). The first concern still seems particularly important to address as it relates to an important claim of the study, and is, in my opinion, in line with another comment from Reviewer 3.

--

1) The authors do not address the concern that the graphs in Fig 2b-c do not show from which replicate the data originate. In line with the comment from Reviewer 3, a super-plot-like representation would allow to display that information on the graph and to exclude the possibility that the wider distribution of entry times of Δ bd1705 is due to only one replicate with slightly longer times (since there is variability between experiments as the authors said). If the effect of the mutation on predation timing is true, the trend (i.e. a wider distribution of entry times for the mutant vs the wt and complemented strain) should be reproducible across experiments even if there is variability. A super-plot would allow to see that nicely (and more rigorously) than by grouping all datapoints, if indeed each experimental replicate included the three strains (WT, mutant, complemented mutant). If I may comment on the authors' response to the comment 2 of Reviewer 3, I respectfully disagree when they say that grouping cells from various experiments (without distinguishing the replicates) is better to reveal phenotypes clearly. Super-plots (which are very simple to obtain) are in fact the best way to ensure that trends are reproducible and that the data grouping does not hide any phenotype or reveal "new" effects not seen in individual experiments (as illustrated in Lord, S.J. et al., 2020. J Cell Biol, 219(6)).

2) While the absence of signal obtained with this new fusion is indeed consistent with a periplasmic localization, this is still an indirect evidence (the fusion might just be unstable for instance). A fusion of mCitrine to the mature Bd1705 (lacking the signal sequence) showing cytoplasmic fluorescence would have been needed as additional control, or a Western blot to confirm the production of the unmodified Bd1705-mCitrine fusion. I suggest that either the authors provide one of these controls or that they rephrase the corresponding lines in the text (e.g. change "was confirmed" by "was consistent with...", "indicating" to "suggesting"

We would like to thank all the reviewers for their time in re-reading our revised manuscript.

Response to reviewers' comments

Reviewer #2 (Remarks to the Author):

Each of my issues have been addressed appropriately/adequately.

We are pleased to hear this. Thank you for your review.

Reviewer #3 (Remarks to the Author):

The reviewers have addressed my concerns and I am highly supportive of publishing this very interesting and well constructed study.

We are glad that we have addressed your concerns and thank you for your kind words regarding our manuscript.

Reviewer #4 (Remarks to the Author) [please see also the attached document]:

I would like to thank the authors for their time and efforts, and for their response to my questions and concerns including new controls and analysis. I have uploaded the rebuttal pdf, in which I have added comments for each of their response (in green). I have two concerns that I could not include fully in the rebuttal pdf file (lack of space), which I am therefore pasting below (I am referring to them in the corresponding response in the file). The first concern still seems particularly important to address as it relates to an important claim of the study, and is, in my opinion, in line with another comment from Reviewer 3.

--

1) The authors do not address the concern that the graphs in Fig 2b-c do not show from which replicate the data originate. In line with the comment from Reviewer 3, a super-plot-like representation would allow to display that information on the graph and to exclude the possibility that the wider distribution of entry times of $\Delta bd1705$ is due to only one replicate with slightly longer times (since there is variability between experiments as the authors said). If the effect of the mutation on predation timing is true, the trend (i.e. a wider distribution of entry times for the mutant vs the wt and complemented strain) should be reproducible across experiments even if there is variability. A super-plot would allow to see that nicely (and more rigorously) than by grouping all datapoints, if indeed each experimental replicate included the three strains (WT, mutant, complemented mutant). If I may comment on the authors' response to the comment 2 of Reviewer 3, I respectfully disagree when they say that grouping cells from various experiments (without distinguishing the replicates) is better to reveal phenotypes clearly. Super-plots (which are very simple to obtain) are in fact the best way to ensure that trends are reproducible and that the data grouping does not hide any phenotype or reveal "new" effects not seen in individual experiments (as illustrated in Lord, S.J. et al., 2020. J Cell Biol, 219(6)).

In the manuscript, we present predator attachment and entry times into prey as box-and-whisker plots which show the full distribution of data from 3 biological repeats. The reviewer has a concern that the data may have been skewed in one direction by just one repeat. This is not the case. 30 cells were analysed per biological repeat and in every biological repeat, the $\Delta bd1075$ mutant invaded prey more slowly than the wild-type, therefore the phenotypic difference was observed **every time**. We would not have presented this data in the manuscript if we had not consistently observed this phenotype.

At the reviewer's request in the original review, we also performed the experiment again with a new complementation strain which completely complemented the entry phenotype. **However, to enable further clarity regarding the provenance of the data, we have amended the legend description of Fig. 2 from "n=90 cells in total from 3 biological repeats" to "n= 90 cells, with 30 cells analysed from each of 3 biological repeats"**.

The reviewer recommends presenting super-plots to show from which repeat the data points originate. **We include super-plots as both bee swarm scatter plots (Supplementary Fig. 8) and violin plots (Supplementary Fig. 9)**. Please note that due to the data points representing discrete minutes (e.g. 4 min, 5 min etc.), many of the data points (90 in total) line up together and are therefore obscured on the bee swarm plot. **We therefore present additional scatter plots as Reviewers' Figures 1 (attachment) and 2 (entry), where the graphs are stretched to ensure that every one of the 90 data points can be clearly visualised.**

In the new **Supplementary Fig. 10**, we further present the data distribution for each of the 3 biological repeats as box-and-whisker plots, showing that the phenotype was observed each time as described above. **We hope that this clarifies the reproducibility of the experiment to the reviewer and enhances the manuscript as they request.**

2) While the absence of signal obtained with this new fusion is indeed consistent with a periplasmic localization, this is still an indirect evidence (the fusion might just be unstable for instance). A fusion of mCitrine to the mature Bd1705 (lacking the signal sequence) showing cytoplasmic fluorescence would have been needed as additional control, or a Western blot to confirm the production of the unmodified Bd1705-mCitrine fusion. I suggest that either the authors provide one of these controls or that they rephrase the corresponding lines in the text (e.g. change "was confirmed" by "was consistent with...", "indicating" to "suggesting")

The sec signal, LD-carboxypeptidase activity on PG and additional Bd1075-mCitrine fluorescence experiment we performed all strongly indicate that Bd1075 is periplasmic. However, at the reviewer's suggestion, we have changed: "Periplasmic localization of Bd1075 was confirmed via" to **"the periplasmic localization of Bd1075 was tested via" (lines 323-324) and "indicating" to "suggesting" (line 326)** to reflect that periplasmic localisation is extremely likely but not completely proven.

Response to Reviewer 4's comments from the separate pdf document (new comments in green)

- Data in Figure 3 do not convincingly support the conclusion about vibrioid shape being a fitness advantage for *B. bacteriovorus*. The frequency of the clear cases of bdelloplast elongation (panel c) should be calculated and compared between wt and $\Delta 1705$ strains. The authors only refer to a "sub-set" at Line 221. Could this sub-set be responsible for the changes in mean bdelloplast dimension shown in panels d-g (mean values being sensitive to low numbers of outliers)?

We have now calculated the percentage of cases of bdelloplast stretching by $\Delta bd1075$ by using a cut-off circularity value of 0.96. We find that 0% of WT bdelloplasts are <0.96 in contrast to 9.2% of $\Delta bd1075$ bdelloplasts which are <0.96 . We now reference this 9.2% being the sub-set of bdelloplasts which deformed by $\Delta bd1075$ in the manuscript at line 185 and describe this calculation in the methods (lines 659-663).

OK, thank you for this analysis. Since this frequency is rather low, I suggest to indicate it on the figure panel as well to avoid the possible misunderstanding that this phenotype is observed for the majority of bdelloplasts.

We have now added this to the figure legend.

I am also not convinced that the statistical significance obtained in panel e (mean \bar{A} SD circularity values: $0.98 \bar{A} \{ 0.04$ vs $0.99 \bar{A} \{ 0.01$) relates to a physiological difference in shape.

The software MicrobeJ (Ducret et al., 2016) was used for all measurements of morphology. Whilst the differences in circularity appear small, this is a result of how this is calculated in the software. This defines circularity as $4\pi \bar{A} \sim \text{area} / \text{perimeter}$. with a value of 1.0 indicating a perfect circle. **To make it clearer that these seemingly small differences are significant, we have added this explanation to the appropriate methods section (lines 659-660) and separate the methods text into two headings: 'phase-contrast and epifluorescence microscopy' and 'image analysis' in order to add more detail to the latter section.** We also attach a reviewers' figure showing a gallery of example cells annotated with their circularity and also curvature values obtained in MicrobeJ. **This also addresses the reviewer's request for an explanation of curvature values further below.** Most circularity values are close to 1.0 – even for rod shapes (often >0.8 A.U.) **therefore a seemingly very small numerical difference in circularity A.U. values e.g. 0.98 vs 0.99 does translate to a meaningful difference in cell shape.**

OK. The authors could include the reviewers' figure in the suppl. information, as this provides support to the significance of these seemingly small changes.

We have added this as Supplementary Fig. 22.

(iii) It is crucial to confirm by western blot that all fusions in all backgrounds are properly produced and not truncated (which would affect the distribution of the fluorescent signal).

At the reviewer's suggestion, we have now carried out western blot analysis using an anti-mCherry antibody for all Bd1075-mCherry fusions (5 fusions in the wild-type background and 5 fusions in the $\Delta bd1075$ background) with appropriate controls. We can see correct production of each Bd1075-mCherry fusion protein at the expected sizes, indicating that each protein is produced correctly. This forms a new figure: Supplementary Fig. 18. **We now include this control in the manuscript text (lines 339-340 and lines 375-376) and have added the experimental methodology to the supplementary methods (lines 131-143 in the Supplementary information).**

Thank you. Could you comment on the much lower amounts of the E302 truncated variant? (in the legend for ex.)

We have commented this in the legend as the reviewer suggests.

- The claim that the vibrioid morphology confers optimal intracellular growth (e.g. line 468) is overstated. The data do not show that the modified bdelloplast shape and the lower curvature of the predator cells impact their growth per se. For this, analysis of growth rates or number of predator cells released per bdelloplast would be needed. The absence of obvious predation efficiency difference between the WT and the $\Delta bd1705$ cells suggests instead that there is no major benefit for growth conferred by curvature. This claim about optimal growth / evolutionary advantage provided by curvature should be softened.

We acknowledge and agree with the reviewer's point and have thus softened the language associated with growth inside bdelloplasts at a number of points throughout the manuscript: the abstract (lines 30-32), introduction (lines 83-85), results (lines 202-203) and discussion (line 523).

Thank you. The title still includes "growth", though.

We think that "growth" is an acceptable substitution for "replication".

- Supplementary Figure 1b:
- please specify at which cell cycle stage were these RNAseq reads acquired.

We have updated the figure legend to specify this.

I do not see the cell cycle stage information, but only the strain (HID13): is this one of the non-predatory mutants? If yes, is the expression profile meaningful considering the authors' previous response about these variants and their "dysregulated gene expression" HID13 is a host-independent mutant grown axenically (but still capable of predation) (Capeness *et al.* 2013). When grown in these conditions, there are simultaneously sub-populations of cells in attack phase and growth phase, so levels of gene expression are dysregulated relative to discrete timepoints in predatory cycles. Therefore this is likely a perfect sample to determine the true transcriptional start site as it represents all different growth conditions. There are no reports of different transcriptional start sites in different samples in *Bdellovibrio*, so again, the determined transcriptional start site is very likely accurate as determined this way.

- Lines 219-220: “despite becoming more curved by the spherical bdelloplast environment over time” is speculative. The authors cannot exclude the existence of other curvature mediating factors in *B. bacteriovorus* during growth, although the confined space seems a plausible cause.

We agree that it is slightly speculative, however, as the reviewer states, the confined space suggests that it is the spherical environment rather than other factors.

OK. I would suggest to rephrase slightly to clarify that this is speculative at this point.

We have changed the language here (line 182) to “despite becoming more curved inside the spherical bdelloplast over time” to reduce the slight speculation of “by the spherical bdelloplast”.

B. bacteriovorus attachment time

Reviewers' Figure 1. Complete scatter plot dataset for *B. bacteriovorus* attachment time to *E. coli* prey as shown in Fig. 2.

B. bacteriovorus attachment time to *E. coli* S17-1 presented as a scatter plot. All data points (90 in total) are shown as a bee swarm scatter plot. Data points are colored red, blue or pink according to the experimental repeat. The median is indicated with 95% confidence intervals. Attachment times between the 3 strains did not significantly differ ($p > 0.05$; Kruskal-Wallis test). 30 cells were analysed from each of 3 biological repeats.

B. bacteriovorus entry time

Reviewers' Figure 2. Complete scatter plot dataset for *B. bacteriovorus* entry time into *E. coli* prey as shown in Fig. 2.

B. bacteriovorus entry time into *E. coli* S17-1 presented as a scatter plot. All data points (90 in total) are shown as a bee swarm scatter plot. Data points are colored red, blue or pink according to the experimental repeat. The median is indicated with 95% confidence intervals. The $\Delta bd1075$ entry time was significantly higher than both WT and $\Delta bd1075$ (comp) ($p < 0.0001$; Kruskal-Wallis test). 30 cells were analysed from each of 3 biological repeats.